# Reduced ATP turnover during hibernation in relaxed skeletal muscle

Cosimo De Napoli[1,2,12], Luisa Schmidt [3,12], Mauro Montesel[1,2], Laura Cussonneau [1,2], Samuele Sanniti[1], Lorenzo Marcucci [2], Elena Germinario[2], Jonas Kindberg [4,5], Alina Lynn Evans[6], Guillemette Gauquelin-Koch[7], Marco Narici[2], Fabrice Bertile [8,9], Etienne Lefai[10], Marcus Krüger [3] ✉, Leonardo Nogara [1,2,11] ✉ & Bert Blaauw [1,2] ✉

Hibernating brown bears, due to a drastic reduction in metabolic rate, show only moderate muscle wasting. Here, we evaluate if ATPase activity of resting skeletal muscle myosin can contribute to this energy sparing. By analyzing single muscle fibers taken from the same bears, either during hibernation or in summer, we find that fibers from hibernating bears have a mild decline in force production and a significant reduction in ATPase activity. Single fiber proteomics, western blotting, and immunohistochemical analyses reveal major remodeling of the mitochondrial proteome during hibernation. Furthermore, using bioinformatical approaches and western blotting we find that phosphorylated myosin light chain, a known stimulator of basal myosin ATPase activity, is decreased in hibernating and disused muscles. These results suggest that skeletal muscle limits energy loss by reducing myosin ATPase activity, indicating a possible role for myosin ATPase activity modulation in multiple muscle wasting conditions.

The regulation of muscle mass and function is affected by changes in activity levels, hormonal stimulation, and mechanical stress[1]. For example, muscle disuse, as occurs when people are bedridden for prolonged periods, or after bone fractures, are accompanied by a significant reduction in muscle mass and function within days/weeks[2]. Surprisingly, despite not eating or moving for months, hibernating bears only lose a moderate amount of muscle mass[3,4], allowing them to look for food after arousal or to get out of the den in an emergency situation[5]. This sparing of body mass is not due to a drastic reduction in body temperature (only a few degrees), as they are able to reduce their basal metabolism independently from body temperature, in part

through a very significant reduction in heart and respiratory rate[6,7]. With regard to skeletal muscle, the mechanisms underlying this reduction in basal metabolism and the preservation of mass and function are not completely understood. Some mechanisms have been proposed, like reduced oxidative stress[8], miRNA-dependent regulation of protein synthesis[9], or circulating factors modulating protein turnover[10], however, no clear mechanism is currently known. As a significant portion of body mass is accounted for by skeletal muscle, reductions in energy consumption by the muscle contractile apparatus can induce major alterations in whole body energy consumption. We have shown that the molecular motor of skeletal muscle, i.e. the

[1]Venetian Institute of Molecular Medicine (VIMM), Padova, Italy. [2]Department of Biomedical Sciences, 35131, University of Padova, Padova, Italy. [3]Institute for Genetics, Cologne Excellence Cluster on Cellular Stress Responses in Aging-Associated Diseases (CECAD), University of Cologne, Cologne, Germany. [4]Norwegian Institute for Nature Research, Trondheim, Norway. [5]Department of Wildlife, Fish and Environmental Studies, Swedish University of Agricultural Sciences, Umeå, Sweden. [6]Department of Forestry and Wildlife Management, Faculty of Applied Ecology and Biotechnology, Inland Norway University of Applied Sciences, Koppang, Norway. [7]French Space Agency, Centre National d'Etudes Spatiales (CNES), Paris, France. [8]Université de Strasbourg, CNRS, IPHC UMR 7178, 7 Strasbourg, Cedex 2, France. [9]National Proteomics Infrastructure, ProFi, Strasbourg, France. [10]Université Clermont Auvergne, INRAE, UNH UMR 1019, CRNH Auvergne, Clermont-Ferrand, France. [11]Department of Pharmaceutical Sciences, 35131, University of Padova, Padova, Italy. [12]These authors contributed equally: Cosimo De Napoli, Luisa Schmidt. ✉e-mail: marcus.krueger@uni-koeln.de; leonardo.nogara@unipd.it; bert.blaauw@unipd.it

**Table 1 | Age, weight and gender of the bears from which biopsies were obtained**

| ID_number | Year of collection | Age (year) | Gender | Summer weight (kg) | Winter weight (kg) |
|---|---|---|---|---|---|
| 1 | 2022 | 3 | F | 47.6 | 49.5 |
| 2 | | 3 | F | 58.2 | 61 |
| 3 | | 3 | F | 72.6 | 67 |
| 4 | | 2 | F | 36.8 | 31 |
| 5 | | 2 | F | 56 | 35 |
| 6 | | 2 | M | NF | 40 |
| 4 | 2023 | 3 | F | 53 | 52 |
| 5 | | 3 | F | 76 | 74 |
| 7 | | 2 | F | 34 | 38 |
| 8 | | 2 | F | 35 | 35 |
| 9 | | 2 | M | 39 | 37 |

Data of bears used in this study showing year of collection, age, gender and weight at capture for each bear. The individual bear number can be followed throughout the manuscript.

myosin protein, can have different ATPase activity based on its conformation in relaxed muscle[11]. It can be found in a biochemical state characterized by very low ATPase activity which is known as the super relaxed state (SRX). On the other hand, if a myosin head is out of the SRX, but still in a relaxed muscle, this has been described as the Disordered Relaxed State (DRX)[12]. These two states have about a tenfold of magnitude difference in energy consumption, as can be measured by ATP hydrolysis rate, with the SRX having a time constant of approximately 200 s$^{-1}$ and the DRX of 20 s$^{-1}$. Many different factors, ranging from changes in pH, temperature, or phosphorylation of myosin regulatory light chains, can modulate the stability of the SRX and alter basal ATPase activity of resting muscle. Interestingly, it was shown to be also modulated in small hibernators[13]. To understand if modulation of myosin ATPase activity in resting muscle contributes to the drastic reduction in whole body metabolic rate in hibernating bears, we analyzed summer and winter freshly skinned biopsies taken from the same bears. Using two different approaches we observed indeed a significant reduction in ATPase activity in muscle fibers taken from winter biopsies. To gain mechanistic insight into the underlying processes regulating this, we performed a single fiber proteomics analysis. This revealed a significant reduction in mitochondrial proteins in winter muscle, confirming results obtained previously on snap frozen muscle[14]. We also observed a decreased myosin light chain kinase activity, possibly contributing to an increased SRX stability, as reported in other experimental systems[15]. Altogether, our results show that part of the energy saving mechanism in hibernating bears is through the reduction of ATP consumption by myosin, likely contributing to whole body energy expenditure.

## Results

### Loss in force production in permeabilized fibers from hibernating bears

Loss of muscle mass and function in hibernating bears is known to be less severe than that observed in humans or mice during muscle disuse. To determine how muscle size and function is affected at the single fiber level, we analyzed muscle biopsies taken from the same bears captured in summer or in winter, around the middle of the denning period. We collected these samples for two years in a row allowing us to analyze 11 winter and 10 summer muscle biopsies. As can be seen in Table 1, bear 4 and 5 were captured for two years in a row, while almost all other bears were captured in one winter and summer. Only bear 6 was not localized again in summer, as young males can migrate to find their own territory when they become young adults. The bears evaluated in this study are not yet sexually mature and have been followed from birth.

To analyze functional properties of muscle fibers, we permeabilized muscle biopsies by placing them in a skinning solution. For this we isolated mechanically from each muscle 8–10 single fibers and determined fiber cross-sectional area. As can be seen in Fig. 1a, there are significant differences in absolute fiber size comparing different bears, however, these can be at least in part explained by the variability in body weight of the different animals. Indeed, bear 2 and 3 (61 and 67 kg) are substantially bigger than number 4 and 6 (31 and 40 kg). Despite these differences in starting weight, average overall fiber size is not changed when comparing summer and winter biopsies. This is in line with the relatively stable body mass between summer and winter for the same bear as shown in Table 1. These cross-sectional areas are obtained on permeabilized fibers which are known to exhibit a 20–30% swelling caused by the chemical permeabilization process. Our results suggest that fiber swelling is not different between summer and winter bears as shown on the right in Fig. 1A. Indeed, the cross-sectional areas of chemically skinned (and swollen) fibers and those measured on snap-frozen muscle sections of the corresponding bears, did not show a significant difference between summer and winter samples (Supplementary Fig. 1b).

Next, we analyzed force production from isolated skinned fibers taken from each biopsy. As shown in Fig. 1b, there was variability in the force production from fibers taken from each individual biopsy. Despite this, we were able to uncover a slight, yet significant 12±2% decrease between summer and winter biopsies, suggesting some alterations in sarcomere contractility during the hibernation period. These observations are in line with those that were reported when measuring twitch force in vivo, showing a 29% reduction after 110 days of denning[16]. It was also reported that twitch kinetics in vivo were reduced during the denning period. To address if part of these alterations in contractile kinetics in vivo can be due to changes in the core contractile apparatus, and not just a consequence of altered calcium handling, we performed a slack test on skinned fibers. This allows us to determine force re-development after a rapid shortening of 10% in an activated fiber, which is sufficient to reduce tension close to zero. In the left side of Fig. 1c two representative force redevelopment traces are reported, showing the slower kinetics of the winter fiber compared to the summer. Force re-development kinetics, expressed as Ktr (Fig. 1c right), are significantly slower in hibernating muscle as compared to control tissue, in line with the observations performed previously in vivo. While it is not trivial to determine which alterations at the sarcomeric level are responsible for these changes in kinetics, it suggests that part of the alterations observed in vivo can be due to changes in the contractile apparatus.

### Hibernating bears reduce ATPase activity of myosin in resting muscle fibers

The lack of food intake and movement, suggests that muscles reduce energy consumption to a minimum. As was postulated more than a decade ago, muscle myosin can be found in two different resting states, the super relaxed state (SRX) and the disordered relaxed state (DRX). Estimates suggest that the SRX, has an ATPase activity which is approximately 5-10-fold lower than that observed when myosin heads are less stable and more disordered (DRX), in both cases without leading to force generation[11]. To determine if relaxed muscle fibers have a lower basal ATPase activity during the winter, we isolated single fibers and performed an ATPase activity assay. This assay is based on two enzymatic reactions that couple ADP produced by myosin-dependent spontaneous nucleotide turnover to NADH oxidation[17]. This assay reflects the activity of myosin, as the myosin inhibitor blebbistatin eliminates most ATPase activity. It is important to point out that these skinned fiber preparations have a highly permeabilized and altered plasma membrane, therefore a lot of ion pumps are no longer functioning. In Fig. 2a on the left, each dot corresponds to the ATPase activity of one single fiber, while the histograms show the

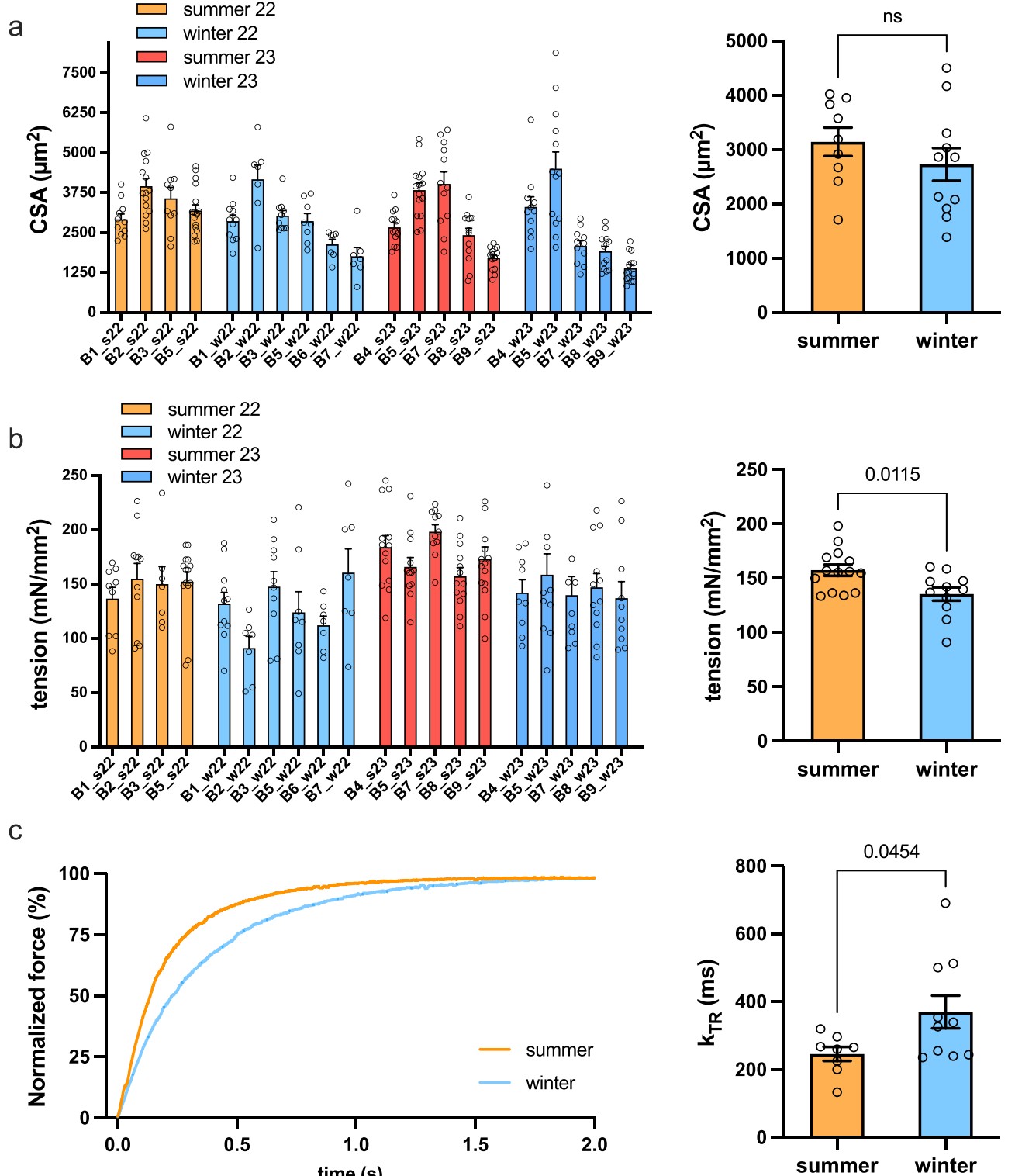

**Fig. 1 | Reduced contractile force and contraction velocity in hibernating muscle fibers. a** Average cross-sectional area of skinned fibers taken from individual bears in winter and summer ($n = 7–17$, Source data are provided as a Source Data file). Each dot corresponds to the CSA of a single fiber taken from that specific bear. On the right the average CSA of all bears examined is divided by season (summer $n = 9$, winter $n = 11$, mean ± SEM, two-sided Unpaired t-test; $P$ value 0.3225). **b** maximal isometric tension produced from skinned fibers taken from individual bears ($n = 7$-15, Source data are provided as a Source Data file) and divided by season summer and winter bears show a small decrease in normalized tension in hibernating bears (summer $n = 14$, winter $n = 11$, mean ± SEM, two-sided Unpaired t-test, $P$ value 0.0115). **c** Representative traces of force redevelopment ($k_{TR}$) after a 10% length shortening in a fiber from a winter biopsy compared to summer biopsy, on the right the average $k_{TR}$ for individual animals (summer $n = 8$, winter $n = 10$. mean ± SEM, two-sided unpaired t-test $P$ value 0.0454). All data related to summer samples are reported in orange/red color, while those related to winter are in light blue/blue.

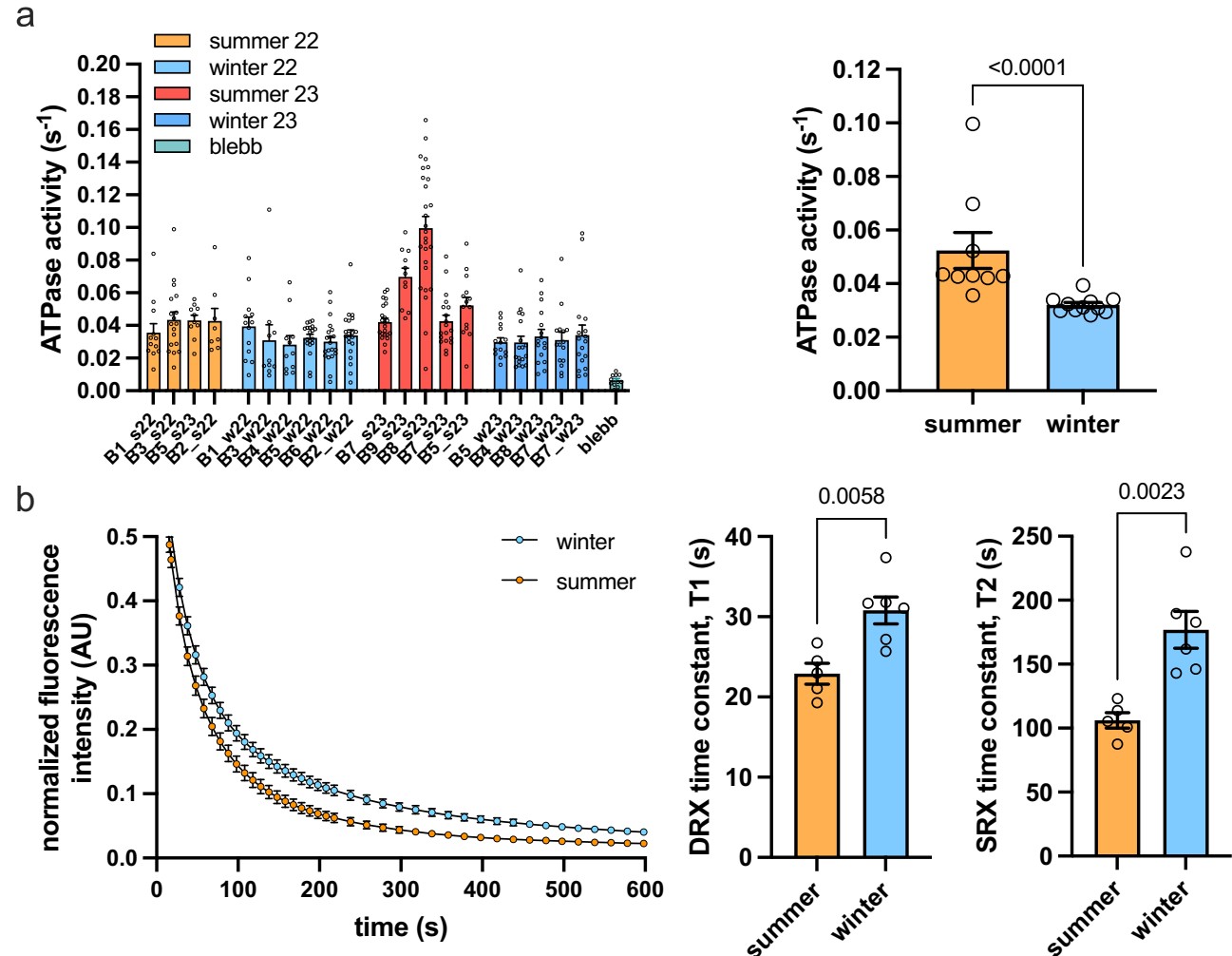

**Fig. 2 | Reduced ATPase activity in hibernating muscle fibers. a** Average ATPase activity of skinned fibers taken from individual bears in winter and summer. Each point represents a single fiber taken from a specific biopsy ($n = 8$-27, Source data are provided as a Source Data file). On the right the average ATPase activity divided by season, in which a decrease in ATPase activity is measured in winter compared to summer (summer $n = 9$, winter $n = 11$. mean ± SEM, two-sided Mann-Whitney test $P$ value < 0.0001). **b** Decaying fluorescence traces of mantATP chasing experiments in skinned fibers from summer and winter bear muscle biopsies ($n = 19$ single fibers for summer samples and $n = 19$ single fibers for winter samples, each fluorescence decay are fitted with the triple exponential decay shown in the material and methods). Fitted parameters are reported on the right, as winter biopsies showed a significant reduction in both DRX (T1. mean ± SEM. Two-sided Unpaired t-test, $P$ value 0.0058) and SRX (T2. mean ± SEM. Two-sided Unpaired t-test, $P$ value 0.0023) time constants, indicating a slower overall nucleotide release. All data related to summer samples are reported in orange/red color, while those related to winter are in light blue/blue.

average for each bear. On the right, each dot corresponds to the average of each individual bear. As shown in b. 2a, basal ATPase rate of bear muscle fibers in summer is $0.046 \pm 0.02 \text{ s}^{-1}$, a range also observed in most other species, like mouse, rabbit and human[18]. Interestingly, fibers taken from the same animals during hibernation show a significant reduction in ATPase rate of $0.033 \pm 0.02 \text{ s}^{-1}$ ($P < 0.0001$). Treatment of fibers with the myosin inhibitor blebbistatin (blebb) reduces ATPase activity to $0.006 \text{ s}^{-1}$ in both summer and winter muscles.

While these results clearly show that energy consumption by resting myosin is reduced in hibernating bear muscles, it does not allow us to determine the relative distributions of SRX/DRX in these fibers. To address this issue, we performed a mantATP chasing experiment, whereby we incubate fibers with a fluorescently labeled form of ATP (mantATP) and determine the decay in fluorescence after changing of the medium. The subsequent observed reduction in fluorescence can be fitted with a triple exponential decay function. The first exponential is very short, since it represents the nonspecific binding of the nucleotide present in solution, the second exponential

is associated to myosin in DRX, while the last one corresponds to myosin in SRX. Each exponential is characterized by two parameters: the population (P) and the time constant (T). Populations are the fraction of myosin heads associated with each state, namely P1 for DRX and P2 for SRX, while the time constant represents the stability of myosin in terms of nucleotide turnover rate. As can be seen in the representative traces shown on the left in Fig. 2b, nucleotide release is slower in hibernating muscles then in summer biopsies, supporting the data presented in Fig. 2a. The altered decay kinetics are caused by significant differences in the time constants T1 and T2, respectively that of myosin DRX and SRX (Fig. 2b, right), being larger in winter and indicating an increased myosin stability. Relative populations of DRX and SRX are not changed in hibernating muscle, as can be observed in Supplementary Fig. 2a.

## Single fiber proteomics shows drastic remodeling in hibernating muscles

How is the proteome affected in hibernating fibers, and how do these changes affect their functional properties? To address these questions,

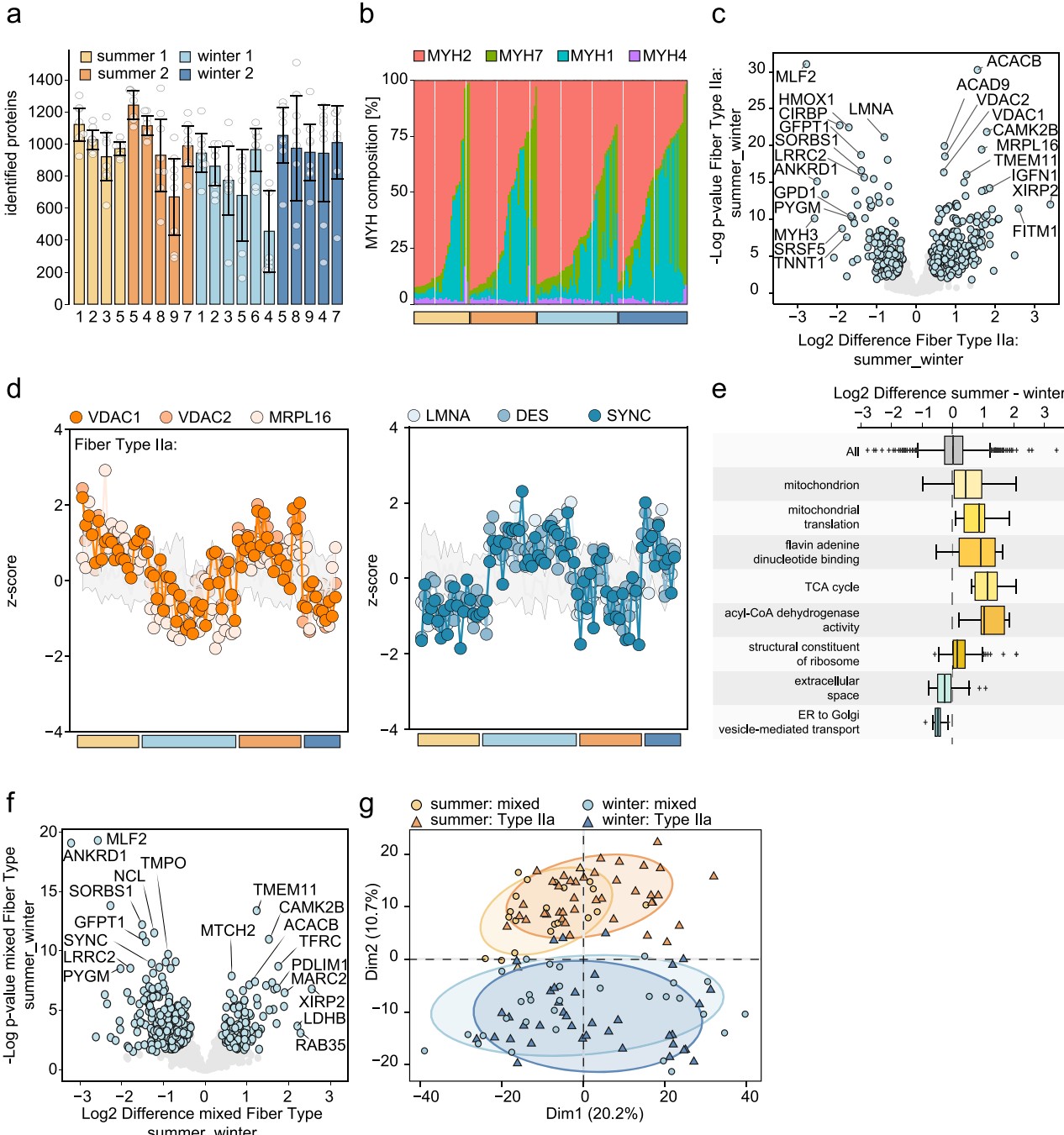

**Fig. 3 | Single fiber proteomics in winter and summer muscles. a** Number of identified proteins per fiber (mean ± SEM; $n = 4$ bears for summer 1, winter 2; $n = 5$ for winter 1, $n = 6$ bears for summer 2) (**b**) MYH isoform distribution analysis of isolated single fibers from different bears over four different seasons (orange summer, blue winter). Each bar represents the relative abundance of MYH isoforms in one fiber in relation to the total sum of intensities for MYH1 (blue), MYH2 (red), MYH4 (purple), and MYH7 (green). Fibers from different bears are sorted into the four seasons and ranked from left to right of highest MYH2 abundance. **c** volcano plot for Type 2A Fibers summer vs winter ($n = 41$–38). Differentially abundant proteins are highlighted in light blue (Welch's t-test: FDR < 0.05, s0 > 0.1) **d** examples of proteins going up in summer (left) or winter (right). **e** changes in GO terms comparing summer and winter type 2A fibers. The box plot represents the upper (75%) and lower (25%) whiskers scores outside the middle 50% with outliers from 0, 100% **f** Volcano plot for mixed (type1/2A) Fibers summer vs winter ($n = 20$-30). Differentially abundant proteins are highlighted in light blue (Welch's t-test: FDR < 0.05, s0 > 0.1). **g** Principal component analysis of protein expression patterns from isolated skinned single fibers of Type 2A (triangle) and mixed fibers (circle) collected in winter (blue) and summer (orange).

we performed a single fiber proteomics analysis on 8–10 single fibers taken from each biopsy. One of the major strengths of this single fiber approach is that one can compare the changes of the proteome within the same fiber type. We conducted our analysis on skinned fibers, which lack many soluble proteins, as our focus was on structural proteins influencing the functional shifts observed. Through this approach, we identified between 700–1200 proteins per fiber (Fig. 3a). HEK cell lysates were analyzed as LC-MS performance control (CV of <5%) and blank controls were injected every 15 samples, identifying minimal protein carry-over per sample (<50) (Supplementary Fig. 3a–c). Using unique peptides, we determined the relative proportion of MYH isoforms in each fiber. Fibers were classified based on

whether they contained more than 70% of a specific MYH isoform. Fiber type distribution remained stable, with Type 2 A fibers being the most dominant in both summer and winter single muscle fibers (Fig. 3b). We confirmed this finding also by an electrophoresis analyses showing how bear MYH isoforms run very similar to those taken from mouse muscles (Supplementary Fig. 2b).

While MYH isoforms showed no major alterations (Fig. 3b), significant changes were observed in the proteome of type 2 A fibers between hibernating and active bear muscle fibers (Fig. 3c, Supplementary Fig. 3d). We identified 402 significantly differentially regulated proteins, with 244 upregulated proteins in summer and 158 in winter. Notable examples like the mitochondrial anion channel (VDAC1/2) are in both summer seasons more abundant while certain myofibrillar components like Laminin-A and Desmin are more abundant in the winter months (Fig. 3d). Fisher exact test performed on the significantly changed proteins showed an enrichment of the gene ontology (GO) terms linked to mitochondrial content and function in summer muscle fibers (Fig. 3e). These comparisons, limited to type 2A fibers, do not reflect shifts in fiber type and concomitant changes in mitochondrial content.

To determine if these changes were specific to type 2A fibers, we also performed a single fiber proteomics analysis on mixed fibers (Fig. 3f, Supplementary Fig. 4). Mixed fibers were defined as those expressing less than 70 % of any MYH isoform. However, they could be categorized as Type 1/2a mixed fibers due to their predominant isoforms being MYH7 and MYH2. Interestingly, many proteins regulated in type 2A fibers showed similar patterns in mixed fibers, indicating a consistent muscle remodeling during hibernation regardless of fiber type (Fig. 3f, Supplementary Fig. 3d). Indeed, principal component analysis demonstrated clear separation between winter and summer single muscle fibers regardless of MYH content (Fig. 3g). ANOVA analysis confirmed clustering of the fibers based on the season rather than fiber types (Supplementary Fig. 4a). Overlapping significantly changed proteins revealed 95 upregulated and 129 down regulated in both fiber type 2 A and mixed fibers (Supplementary Fig. 4b, c).

To better understand how these changes in mitochondrial GO terms reflect their organization and content within the fibers, we first performed a western blotting analysis for different proteins of the five respiratory complexes. In line with the observations in the single fiber proteomics, we observed a small, yet significant reduction in specific proteins from each complex (Fig. 4a). Next, we performed an immunohistochemistry analysis for Tom20, a mitochondrial import receptor subunit, to determine mitochondrial distribution within the fiber. As can be seen in Fig. 4c, summer fibers showed a more intense staining with a relatively normal distribution pattern. Electron microscope analyses of summer and winter skinned fibers showed, as expected, increased spaces between sarcomeres and swollen mitochondria in both conditions, as compared to freshly fixed tissue (Supplementary Fig. 4d). More in general, a schematic representation of differentially regulated mitochondrial proteins shows how proteins linked to lipid oxidation are reduced, while glycolytic proteins are maintained, similar to previous observations[14] (Fig. 4d).

### Hibernating muscles shows a downregulation of myosin light chain kinase content and activity

As mentioned, there are 244 proteins with a significantly higher expression in summer muscles. To better understand what these proteins have in common, we performed an analysis using the ENRICHR software. One of the most suggestive enrichments we observed in this list is related to the so-called kinases co-expression analysis. In this analysis it is possible to determine the kinases which are co-expressed with regards to the list of proteins/genes examined. Interestingly, we observed that MYLK2 is the kinase with the strongest correlation to the enriched proteins in the summer muscles (Fig. 5a). This captured our interest, as a reduced MYLK2 expression or activity

during the winter induces a decrease in myosin regulatory light chain (RLC) phosphorylation, with subsequent stability of the SRX and a reduced ATPase activity. As skinned fibers are permeabilized, they do not allow for the determination of changes in phosphorylation levels. Therefore, we used snap frozen muscle tissue from the same bears and performed a western blotting analysis for both the total kinase levels and the phosphorylation of its main target, the myosin regulatory light chain (MLC). As shown in Fig. 5b and 5c, we find a very consistent decrease in both MKL2 content and the phosphorylation of RLC on serine 19, the site known to be involved in the regulation of SRX stability. As it is not straightforward to find a specific protein which is unaltered during hibernation, blots were normalized for total protein content as identified by ponceau staining. Interestingly, this reduction in MYLK2 is also in line with a transcriptional reduction observed in a previous study from hibernating bears[19], or during torpor in zebrafish[20]. To understand if a similar reduction in MYLK2 levels also occurs in other models of muscle disuse, we analyzed its expression levels in a recently developed murine model of unilateral hindlimb casting[21]. As can be seen in Fig. 5d and e, there is a significant reduction in MYLK2 after 7 and 14 days, suggesting a preserved mechanism between mice and bears during muscle unloading.

Taken together, our results suggest that part of the reduced ATPase activity in winter muscle is a consequence of an increased SRX stability due to reduced RLC phosphorylation.

## Discussion

It is well known that muscle disuse leads to a very rapid loss in skeletal muscle mass and function in a matter of days. In this sense, hibernation represents a unique form of disuse, during which animals abstain from eating or engaging in physical activity for months, yet remarkably, experience only minimal muscle loss. For comparison, a 90-days immobilization in humans leads to a decrease in force of 54%[22]. On average, the effect of bed-rest and immobilization periods of 4–28 days showed a muscle loss in the range of 0.2–2.3 %/day, which is always exceeded by a force loss, ranging between 1.1–3.5 %/day[2,23]. In a disuse model in rodents caused by 3 d printed cast, the gastrocnemius muscle exhibits a 25% in weight loss and a 40% drop in muscle force over the course of 2 weeks[21]. Also in this study, despite not measuring muscle mass or function in vivo, we find no atrophy in single fibers and only a 12% reduction in force-generating capacity, suggesting significant protection against disuse muscle wasting.

It has been reported that this remarkable tissue sparing during hibernation is achieved, among other mechanisms, by drastically reducing metabolic rate[3,6]. Indeed heart rate and respiratory rate are reduced 3–5 fold, despite maintaining body temperatures relatively high[6]. Here, we determined if skeletal muscle myosin, one of the most abundant proteins in the body, has a reduced ATPase activity in hibernating bears. Using a single fiber ATPase assay, we find that ATPase activity in relaxed skinned fibers of hibernating muscle is significantly lower (-28%) than what is observed in muscle biopsies taken from the same bears in summer. Using a mantATP chasing approach we find that time constant for nucleotide exchange is prolonged for all myosin heads in winter samples. Performing a single fiber proteomics analysis, we find, despite no major loss in fiber size and function during hibernation, that there is a very significant remodeling of the proteome. Mitochondrial proteins are altered and show a reduction in winter muscle, independently from fiber type. Interestingly, winter fibers show a lower content and activity of MYLK2, a kinase responsible for myosin RLC phosphorylation which induces SRX destabilization, possibly explaining the different resting ATPase activity in the two seasons.

### Muscles from hibernating bears show altered contractile properties and protein content

In a landmark paper published over 20 years ago, it was shown that hibernating bears only lose around 20% of muscle strength in vivo

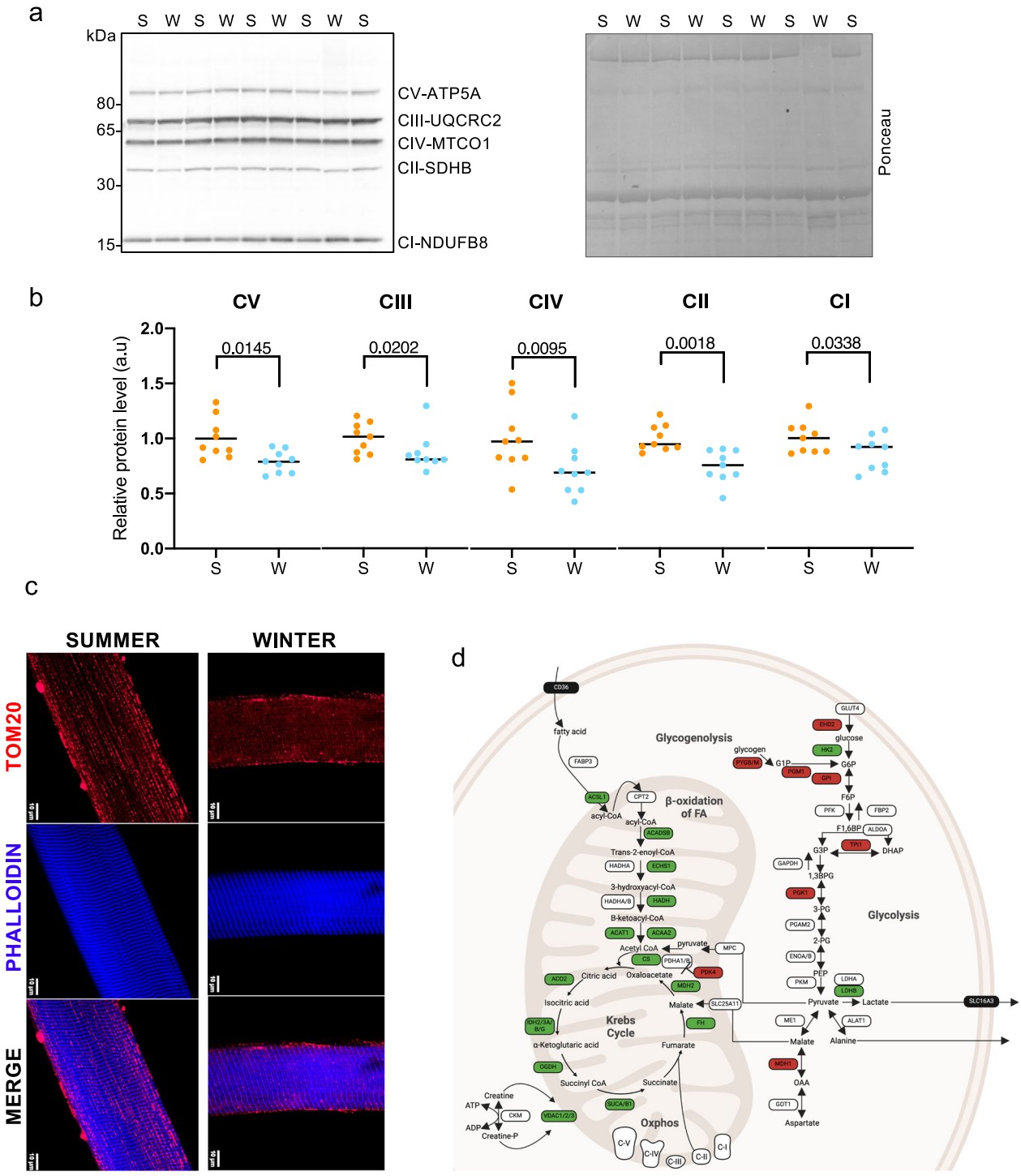

**Fig. 4 | Reduction in mitochondrial content during hibernation. a, b** Western blotting for proteins of the different respiratory complexes shows a significant reduction in all complexes in winter (W, blue dots) compared to summer (S, orange dots) skinned fibers. Data are presented as individuals' values with mean bars ± SEM (*n* = 9 bears/season, the same individuals were sampled and analyzed in summer and winter) and normalized to the total protein content (Ratio paired t test two-sided, CV *P* value = 0.0145; CIII *P* value = 0.0202; CIV *P* value = 0.0095; CII *P* value = 0.0018; CI *P* value = 0.0338; a.u=arbitrary units). Representative Western blots are shown for four couples of bears. Samples are derived from the same experiment and blots were processed in parallel. Source data are provided in the Source Data file **c**. Immunohistochemistry for TOM20 shows mitochondrial distribution, while Phalloidin shows actin localization in summer and winter fibers (*n* = 3 fibers for animal, *n* = 3 bears for season. The same individuals were used for summer and winter fibers). **d** Regulation of metabolism-related factors in skinned fibers of hibernating bear muscles. The relative abundance of proteins in winter (hibernating) versus summer (active) brown bears (*n* = 6 per season) is shown using the following color code: significantly (Welch T-test analysis; *P* < 0.01) up- and down-regulated proteins are shown in red and green boxes, respectively; white boxes show proteins that were unchanged between winter and summer and black boxes show proteins that could not be detected. Created in BioRender. Cussonneau, L. (2024) https://BioRender.com/t32y880.

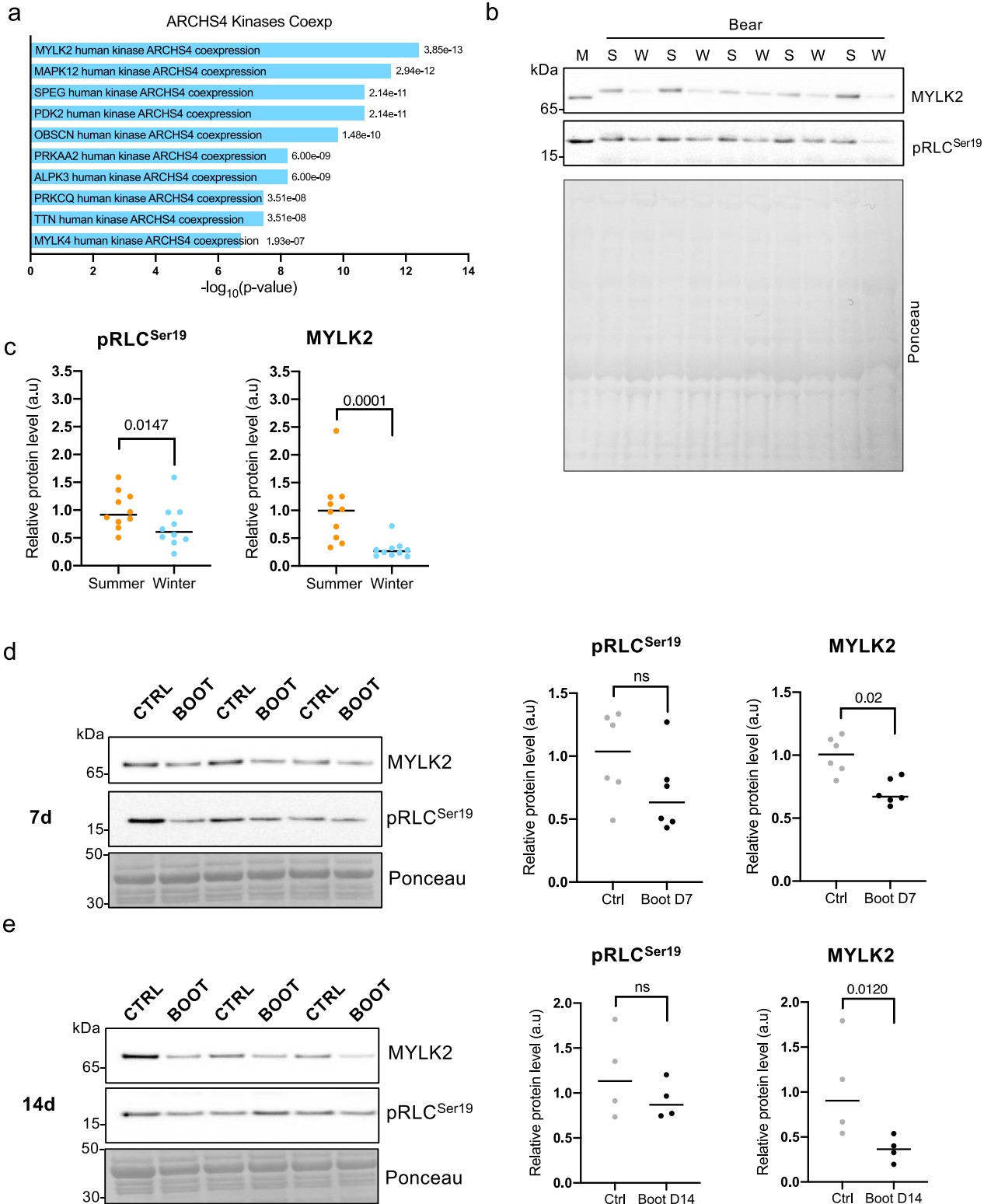

during hibernation[24]. Later, the same authors confirmed these observations and also showed that contractile kinetics of twitch tension was reduced[16]. As these measurements were performed in vivo, it is difficult to determine where muscle dysfunction or changes in kinetics occurred. Muscle disuse is known to lead to alterations in muscle innervation, calcium dynamics and dysfunction of the contractile apparatus[2,25,26]. In our study, we evaluated contractile function and kinetics of single fibers, without the requirement of muscle innervation or calcium release through normal excitation-contraction coupling

cycles. Interestingly, we find that most of the muscle dysfunction which occurs in hibernating bears is due to a reduced force production from the contractile apparatus. A possible explanation for a reduced normalized force in single fibers could potentially be through a different fiber swelling[27], as reduced swelling in winter fibers would lead to an underestimation of skinned fiber cross sectional area. However, as winter fibers show an increase in structural proteins like desmin and lamin A, likely reinforcing the cytoskeleton, it is unlikely winter fibers swell more. Indeed, comparisons of snap frozen and skinned CSA do

**Fig. 5 | Reduced myosin light chain kinase activity in disused muscle. a** ENRICHR kinases co-expression analysis shows an increase in MYLK2 in summer bears. **b, c** Western blotting analysis on frozen muscle tissue taken from the same bears as the skinned fibers shows decrease in MYLK2 and the phosphorylation of its target, P-RLC. Data are presented as individuals' values with mean bars ± SEM (n = 10 bears/season, the same individuals were sampled and analyzed in summer and winter) and normalized to the total protein content. Orange and blue dots are for bear muscles, respectively in summer and winter (Ratio paired t-test two-sided, pRLC P value = 0.0147, and MYLK2 P value = 0.0001; a.u.=arbitrary units). Representative Western blots are shown for five couples of bears and a control mouse muscle (M). **d, e** Western blotting analysis on gastrocnemius muscle tissue from immobilized (Boot D7 or D14, black dots) or the controlateral muscle (Ctrl, grey dots) from the same mouse show a significant decrease in MYLK2 and a tendency for the phosphorylation of its target P-MLC in immobilized leg compared to the contralateral one. Data are presented as individuals' values with mean bars ± SEM (n = 6 mice for one week, D7 and n = 4 mice for 2 weeks, D14 immobilization) and normalized to the total protein content. Ratio paired t test two-sided (pRLC D7, P value = 0.1822; MYLK2 D7, P value = 0.02; pRLC D14, P value = 0.2511; MYLK2 D14, P value = 0.0120; a.u = arbitrary units). Representative Western blots are shown for 3 couples of mice. Samples are derived from the same experiment and blots were processed in parallel. Source data are provided as a Source Data file.

not show differences between summer and winter fibers. Also, electron microscopy analysis does not show increased space between sarcomere in parallel or the presence of damaged sarcomeres. This suggests that reduced force is most likely due to alterations in the force generating capacity of the contractile proteins in the fiber. One of the few modifications known to affect functional properties of skinned fibers is oxidation, leading to reduced contractile force and kinetics[28]. Indeed, incubation of permeabilized fibers with a nitric oxide donor, is sufficient to reduce contractile force and kinetics, but even resting ATPase activity, suggesting this oxidative stress acts on the myosin head and not on the actin-myosin interaction. It has been reported that antioxidant defense and general oxidative stress in hibernating muscles is reduced, even though overall nitrosylation is increased[8]. However, it is possible that proteins with a very high half-life, like structural muscle proteins[26], can accumulate these relatively stable oxidative modifications over time, leading to the observed functional decrease. This issue is likely to be particularly pronounced during hibernation, as it has been reported that general protein turnover is strongly reduced[4].

While we were not able to link the functional deficit to changes in a specific protein, the proteomics analyses performed on single fibers show a major remodeling of the proteome in both fast and slow fibers. The most obvious alteration is observed in the mitochondrial proteome, with many proteins showing a significant reduction. Similar results have been reported in many hibernating species[29], and could contribute to some redox-dependent modifications of contractile proteins. While many proteins show a decrease in winter muscle, there are also interesting proteins which strongly increase during hibernation that can potentially explain some of the protective processes that occur under these extreme conditions. One interesting example is Cold-inducible RNA binding protein (CIRBP), an mRNA binding protein, which can stabilize specific transcripts to improve survival under prolonged cold conditions. Indeed, loss of CIRBP reduces hypothermic cardio protection[30], a critical issue for heart transplant. Interestingly, this protective effect of CIRBP appears to be mediated by its inhibitory effect on ferroptosis[31], a specific type of cell death accompanied by iron accumulation and lipid peroxidation[32]. While the regulation of ferroptosis as a protective mechanism during hibernation has been suggested[33], it is interesting to note that similar changes might also occur in hibernators which do not lower their body temperature that much, like the bears.

### Hibernating muscles reduce myosin ATP consumption

The major finding in this study is that relaxed skeletal muscle myosin consumes less ATP during hibernation. Both an ATPase activity measurement (NADH-oxidation) and a MantATP chasing approach clearly showed that myosin in single muscle fibers taken from hibernating bears consumes less ATP. While summer fibers showed some variability, winter fibers had a consistently reduced ATPase activity in all biopsies examined. Some of this variability observed in the ATPase activity in the summer group can be due to the high variability in activity patterns and daily energy expenditure reported in bears[34]. In non-denning adult polar bears daily energy expenditure can vary 5-10-

fold, while activity patterns can show an even wider range. The more homogeneous values obtained in winter muscles are most likely due to the more uniform conditions at the moment all bears are captured. Changes in ATPase activity and ATP/ADP levels are also known to affect contractile kinetics, linking metabolic changes directly to contractile characteristics. Indeed, using an in vitro motility assay it was shown that sliding velocity of myosin is strongly affected by ADP dissociation from actomyosin and decreased in dystrophic muscle, a condition known to have an increased oxidative stress[35,36]. A reduction in ATPase activity of myosin is also supported by findings from snap frozen biopsies that show a reduction in both the production and use of ATP in the muscles of hibernating bears[14]. Proteomics studies suggest that this is accompanied by a preferred use of lipids, albeit with a lower rate of oxidation, while sparing glycogen stores[14]. Furthermore, in a recent study it was found that the ATPase activity in small hibernators was altered, however, no evidence for changes in bear muscle was observed[13]. While this might seem in contrast to our results, we believe that an important reason for the discrepancy is that our samples were skinned starting with fresh material, without experiencing snap freezing. It has been shown that the freezing/thawing process reduces force production by roughly 50%[37–39], likely altering the ordered pattern of thick filaments, which has an important role in the SRX cooperative nature.

MYLK2 activity through RLC phosphorylation is a well-known mechanism that causes an increased calcium sensitivity and an altered thick filament conformation, with more disordered myosin heads arrays[40]. In response to this phosphorylation, an increase in resting myosin ATP consumption has been reported for cardiac myofibrils[41], while its energetic contribution in skeletal muscle is controversial, with some literature stating little-to-no effect[42] and others suggesting an increase only at low calcium levels[43]. It must be stated that divergent outcomes may be due to an undetermined initial sample phosphorylation level, as well as the marginal modulation of nucleotide turnover in a resting muscle compared to active contraction. In fact, myosin light chain phosphorylation does not alter the stability of the SRX in a linear fashion and does not abolish it completely[11]. MYLK2 is rapidly activated by the calmodulin-calcium complex in a contracting muscle, while it is slowly deactivated by its autoinhibitory alpha helix in absence of free cytosolic calcium. In skeletal muscle, the phosphatase kinetics are quite slow[44], leaving myosin RLC phosphorylation as a molecular memory of muscle activation, and inducing post tetanic potentiation[45]. As muscles are highly inactive during hibernation, it is reasonable to assume that the lack of calcium release from the SR is one of the major reasons for the reduced MYLK2 activity, even though changes in phosphatase activity cannot be excluded either for the reduced RLC phosphorylation.

According to our data, the decrease in ATP consumption from myosin during hibernation is caused by a prolongation of myosin DRX and SRX time constants, as shown by the mantATP chasing experiments. Even in the absence of a change in DRX and SRX populations, an increased time constant reflects a different myosin heavy chain heads stability. In fact, in skinned fibers from winter biopsies the DRX constant (T1) is increased by about 25% while the SRX constant (T2) is

almost double. The corresponding difference in nucleotide exchange rate causes the increased stability of myosin in the DRX population to contribute in a stronger manner to the decrease in ATP hydrolysis, followed by the increased stability of myosin in the SRX population. These data support the idea that an energy preservation mechanism is acting on the biochemical equilibrium of the thick filament. It has been reported that age could affect myosin SRX in the same fashion we showed here[46], by altering time constant rather than myosin populations. It is known that, upon oxidation, myosin produces less force, but its effect on nucleotide exchange and possibly on myosin heads dragging on the thin filament in a lightly attached configuration is not well established. Temperature is unlikely to have a major impact on the seasonal differences reported in this work. Indeed, bears during hibernation do not experience a massive drop in body temperature, not enough to induce a structural destabilization and the appearance of myosin heads in the refractory state[47]. Also, it is known that with the lowering of the temperature myosin heads undergo an order-to-disorder transition[48], increasing DRX and consequentially futile ATP hydrolysis. To give an estimation of the contribution of the SRX/DRX balance to whole body energy consumption, we performed a rough calculation based the energy consumption of resting muscle myosin (Supplementary Fig. 5). These suggest that energy sparing in muscle tissue is around 30% in hibernating muscle compared to summer muscle.

Taken together, we find that in large, hibernating mammals, like the brown bear, skeletal muscle myosin reduces ATPase activity, highlighting a new energy saving mechanism during extreme situations. Interestingly, also a murine model of muscle disuse showed a similar reduction in MYLK2 levels. While additional mechanistic exploration is required to link MYLK2 levels to resting energy consumption in muscle fibers, it does suggest that myosin ATPase activity might also play a role in other conditions of muscle wasting, particularly those affected by major modulations of muscle metabolism.

## Methods

### Bear Sample Collection
Biopsies from the vastus lateralis muscle were collected from 9 free-ranging brown bears, 2–3 years old (Ursus arctos), from Dalarna and Gävleborg counties, Sweden, from 2022 to 2023 (Table 1). The samples were immediately frozen on dry ice until storage at −80 °C or processed to obtain a chemically skinned sample. Each year, the same bears were captured during winter hibernation (February) and recaptured during their active period by helicopter darting[49] (June). Two bears were captured in two consecutive years. Bear immobilization was performed as previously described[50]. The study was conducted according to the guidelines of the Declaration of Helsinki and of the European Directive 2010/63/EU and approved by the Institutional Review Board (or Ethics Committee) of (1) the Swedish Ethical Committee on Animal Experiment (Dnr 5.8.18-03376/2020), the Swedish Environmental Protection Agency (NV-01278-22), the ARRIVE guidelines, and the Swedish Board of Agriculture (Dnr 5.2.18−3060/17). All procedures complied with Swedish laws and regulations. Capture, anesthesia, and sampling were carried out according to an established biomedical protocol[51]. For skinned fibers, bundles of bear muscle were harvested and stored at 4 °C in a skinning solution (150 mM potassium propionate, 5 mM potassium dihydrogen phosphate, 5 mM magnesium acetate, 5 mM EGTA, 2 mM DTT, 2.9 mM ATP, 0.5 mM sodium azide, proteases and phosphatases inhibitors, pH 7.0) for 24 h and then transferred to a storage solution (same as skinning but 50% glycerol) at -20 °C[17,52].

### Mouse Sample Collection
Mice were housed in independent cages in an environmentally controlled room (23 °C, 12-h-light-dark cycle) with ad libitum access to food and water. Gastrocnemius muscles of 3-month-old C57BL/6 mice were collected at different time points: after 1 week (D7, $n = 6$ mice) and 2 weeks (D14, $n = 4$ mice) of unilateral immobilization using custom-made 3D-printed boot as previously described[21]. Sex has not been considered in this study, both male and female mice have been included. Gastrocnemius muscles of the controlateral leg were collected at the same time and all the muscles were snap-frozen in liquid nitrogen for subsequent analyses. The study was conducted according to the Guide for the Care and Use of Laboratory Animals (NIH; National Academies Press, 2011), to the ARRIVE guidelines (https://arriveguidelines.org/) as well as the Italian law for the welfare of animals. The Italian Ministero della Salute approved all animal experiments, Allegato VI (Rome, Italy; authorization number 448/2021 PR).

### Western Blot
Vastus lateralis muscles or skinned fibers from 9 bears (paired samples collected in summer and winter each year for the same individual; Table 1), and gastrocnemius muscles from mice after 1 week (D7, $n = 6$) or 2 weeks (D14, $n = 4$) of unilateral immobilization or the controlateral leg (Ctrl), were used. Samples were homogenized using metallic beads in 100 μL of an ice-cold buffer (50 mM Tris pH 7.5, 150 mM NaCl, 1 mM EDTA, 10 mM MgCl2, 0.5 mM DTT, 10% Glycerol, 2% SDS) containing inhibitors of proteases (complete Tablets EDTA-free, Roche) and phosphatases (phosphoSTOP, Roche, Monza, Italy). The homogenates were lysed for 3 min at 30 Hz in TissueLyser II (Qiagen, Milan, Italy) and then centrifuged at $10,000\,g$ for 15 min at 4 ∘C. The resulting supernatants were then stored at −80 °C until further use. The concentration of proteins was determined using the Pierce™ BCA Protein Assay (Thermo Fisher Scientific, Parma, Italy). Proteins were then diluted in Laemmli buffer and stored at −80 °C until further use. Protein extracts were subjected to SDS-PAGE (sodium dodecyl sulfate-polyacrylamide gel electrophoresis) using Bolt™ 4 to 12%, Bis-Tris Plus WedgeWell™ gels (Thermo Fisher Scientific, Parme, Italy) and transferred onto a nitrocellulose membrane. Blots were blocked for 1 h at room temperature with 5% bovine serum albumin in TBS buffer with 0.1% Tween-20 (TBS-T, pH 7.8), then washed thrice in TBS-T and incubated (stirring overnight at 4 °C) with appropriate primary antibodies against Phospho-Myosin Light Chain 2 (Ser19)(Rabbit, 1:1000 5%BSA, CST#3671, Lot: 7, Cell Signaling Technology, Milan, Italy), MYLK2 (Rabbit, 1:1000 5%BSA, HPA059704, Lot: R84232, Atlas antibodies, Rome, Italy) and OXPHOS (Mouse, 1:2000 5%BSA, #ab110413, Lot: 2101006616, Abcam, Milan, Italy). Blots were washed and incubated for 1 h with an appropriate secondary horseradish peroxidase-conjugated antibody at room temperature, goat Anti-Mouse IgG-HRP Conjugate (1:5000) and Goat Anti-Rabbit IgG-HRP Conjugate (1:4000), respectively #1706516 (Lot: L005680) and #1706515 (Lot: L005679), Biorad, Segrate MI, Italy. Signals were detected after incubation with Luminata Classico Western HRP substrate (Millipore, Burlington, MA, USA) and visualized using iBright750 imaging system (Thermo Fisher Scientific, Parme, Italy). Signals were quantified using the ImageJ software 1.53f51[51] and normalized against the total amount of proteins determined by Ponceau signals to correct for uneven loading. Protein data were presented as individual values. The bilateral ratio paired Student's t-test was used to compare the muscles of bears during summer and winter (S and W, respectively). Statistical analysis was performed using Prism 8 (GraphPad Prism 9, San Diego, CA, USA).

Muscle samples for protein electrophoresis were solubilized in the SDS-PAGE sample buffer (62.5 mM Tris pH 6.8, 2.3% SDS, 5% Beta-mercaptoethanol,10% glycerol) containing the Complete Protease Inhibitor Cocktail (Roche, Basel, Swiss) and analyzed by SDS-PAGE on 8% polyacrylamide gels according to the method described by Talmadge and Roy[52]. MyHC protein bands were revealed by Coomassie Blue (EZBlue Gel Staining Reagent, Sigma Aldrich) and isoform percentage composition was evaluated by densitometry using ImageJ.

## mantATP chasing

Single fibers were dissected from biopsies and then mounted on aluminum T-clips on a 3D printed setup (Supplementary Fig. 1A). The setup was then placed on the stage of a Nikon Eclipse inverted fluorescence microscope equipped with a Hamamatsu ORCA Flash-4.0 camera. The sarcomere length was adjusted between 2.4–2.5μm using a micromanipulator. The experiment was performed at room temperature (26 °C). The fiber was incubated in a rigor buffer solution (potassium acetate 120 mM, MOPS 50 mM, EGTA 4 mM, potassium dihydrogen phosphate 5 mM, magnesium acetate 5 mM, DTT 1 mM, pH 6.8) for 5 min. Then, the fiber was moved to another chamber containing rigor buffer plus 250 μM mantATP (NU-202L, Jena Bioscience, Germany) and incubated for 10 min. At the beginning of the recording, the fiber was moved to a new chamber containing fresh relaxing buffer (rigor buffer with the addition of 4 mM ATP). Pictures were taken using the DAPI filters set through a 10X magnification objective (PLAN FLUOR 10X/0.30 WD 16.0), plus additional 1.5X magnification of the Nikon Eclipse stage. The recording protocol was the following: 10 frames were taken every 2 s, then 20 frames every 10 s, and 20 additional frames every 20 s, for a total measurement time of 620 s. Every frame exposure was 400 ms with the camera set to binning 4×4. The recording protocol and camera setting has been optimized to reduce photobleaching during the total illumination time of about 50 s. The decrease in fluorescence intensity was fitted with the following three exponential decay function:

$$Y = 1 - P0^* \left(1 - 1^{(-x/T0)}\right) - P1^* \left(1 - 1^{(-x/T1)}\right) - P2^* \left(1 - 1^{(-x/T2)}\right)$$

where P0, P1 and P2 are population intensities, while T0, T1 and T2 are population time constants. Initial fitting values are set to P0 = 0.25, T0 = 2, P1 = 0.375, T1 = 20, P2 = 0.375, T2 = 200 and fitting is run for 1000 iterations using GraphPad Prism (version 10.0.2 for Mac, GraphPad Software, Boston, Massachusetts USA). P1 and P2 are expressed as a percentage of P1 + P2, thus the total fitted populations, excluding the non-specific P0.

**NADH coupled reaction ATPase activity assay.** Single fibers were dissected and pipetted into a 384well plate in relaxing buffer (Potassium propionate 100 mM, MOPS 50 mM, EGTA 12 mM, magnesium acetate 6 mM, potassium dihydrogen phosphate 6 mM, ATP 4 mM, DTT 2 mM, Triton X-100 0.025%, protease inhibitor, pH 7.4) plus the coupled reaction buffer (NADH 1.6 mM, PEP 5 mM, pyruvate kinase 40U/ml and lactate dehydrogenase 40 U/mL) to a final volume of 30 μL. The plate was covered with an optically clear seal and placed in a temperature-controlled multiplate reader set to 27 °C (Multiskan SkyHigh, ThermoFisher Scientific). The coupled reaction act as the following: myosin hydrolyses ATP to ADP and inorganic phosphate, ADP and phosphoenolpyruvate are converted by pyruvate kinase to ATP and pyruvate, pyruvate and NADH are converted to lactate and NAD+ by Lactate dehydrogenase[17,53]. The oxidation rate of NADH was measured every 2 min as the decreasing absorbance at 340 nm, total run time 25 min. The linear decrease was estimated using NADH absorbance epsilon of 6300 mol$^{-1}$ cm$^{-1}$ and using a calibration curve obtained by the addition of a growing concentration of ADP. A concentrated KCl solution was added to each well at the end of the assay to reach the final concentration of 0.4 M and extract myosin from the bear fiber, the protein amount was measured using Pierce™ 660 nm Protein Assay Reagent (Thermofisher nr. 22660) for normalization purposes.

**Single fibers tension measurement.** Single skeletal muscle fibers were dissected in cold storage solution, clipped with aluminum T-clips at each end and mounted on an Aurora Scientific Permeabilized Fibers 802D setup (Aurora Scientific, Ontario, Canada). A thin flow of 5% toluidine blue and 8% glutaraldehyde[54] was used to crosslink each end

of the fiber. After an extensive wash with relaxing solution, the diameter was measured, and sarcomere length was adjusted to 2.5 μm using a camera mounted on an inverted microscope. The maximal tension of the fibers was measured at 21°C. For single fiber shortening the sarcomere length was adjusted to 2.9 μm, so during the activation of the fiber a steady shortening of 10% $L_0$ was imposed at the plateau. Relaxing, preactivating and activating buffers, as well as calcium sensitivity buffers are derived from[48]. Force is normalized to cross sectional area using the measured diameter and assuming circular geometry. The time constant of the force redevelopment after a slack test has been obtained through a single exponential fitting. The tension-time trace has been analyzed between the time of the initial rise in tension just above the zero level, reached after a shortening of 10% of the SL, to the time when the plateau was fully reached. The build-in MATLAB® function fit has been used with the custom function $T = a(1 - e^{-\frac{t}{t_c}})$, being T the tension, t the time, a the tension at the plateau and $t_c$ the time constant. R2 values were higher than 0.9 for all the traces.

Ktr is described as the rate of force redevelopment following a rapid shortening of the fiber after reaching maximal isometric tension. The rapid shortening allows the tension to be redeveloped from zero in a saturating calcium condition so that the kinetic is not affected by Ca2+ diffusion, the thin filament is activated, and the force increases proportionally to the thick filament activation and myosin recruitment. The experimental traces are fitted using a single exponential function as reported extensively in the literature[55].

## Single fiber proteomics

Isolated single fibers ($n = 8$) of $n = 4$ (summer 1, winter 2), $n = 5$ (winter 1), $n = 6$ (summer 2) biological replicates per season were placed in a 96-well plate. On each plate, one column was used for the quality standard based on 20k HEK cells to process sample preparation and the performance of the LC-MS instrumentation. 40 μl of 4% SDS in PBS containing 5 mM TCEP and 10 mM CAA were added to each well. Next, 96 well plates were placed on 95 °C for 10 min followed by sonication in a Bioraptor sonicator set to 20 °C water temperature with 10 cycles of 30 on/30 off. Samples were digested following the standard SP3 protocol[56]. Briefly, 20 μg of each washed SP3 beads were added to each well followed by immediately adding one sample volume of acetonitrile (ACN). After the incubation period of 8 min and 2 min on a magnet, supernatant was discarded and magnetic beads were washed 2x with 70% EtOH, 1x with 100% ACN. Samples were digested with 10 μl of 20 ng LysC and 40 ng trypsin dissolved in 50 mM ammonium bicarbonate at 37 °C, 750 rpm overnight. Samples were acidified by using 100 μl 0.1% FA followed by a clean-up with house-made SDB-RPS tips. Purified peptides were loaded with indexed retention time peptides (iRT) on EvoTips Pure and applied a 60 SPD method on an EvoSep One system (both EvoSep, Denmark) with an 8 cm PepSep Column. Mobile phases were compromised of 0.1% FA as solvent A and 0.1% FA in ACN as solvent B. The HPLC system was coupled to a timsTOF pro 2 using a CaptiveSpray source (both Bruker). Samples were measured randomly in dia-PASEF mode with daily ion mobility calibration using three ions of Agilent ESI-Low Tuning Mix following vendor specifications. The DIA-PASEF window was ranging in dimension 1/k0 0.7–1.35, with 24 × 25 Th windows and in dimension m/z from 350 to 1250. The mass spectrometry proteomics data have been deposited to the ProteomeXchange Consortium via the PRIDE partner repository with the dataset identifier PXD050980[57]. Files were processed with DIA-NN 1.8.1 using library free search against UniProt Ursus Arctos database (2019) complemented with protein sequences from myosin heavy chain variants. Database was blasted against existing GOterms with GOblast. Mass ranges were set according to the settings of the mass spectrometer, mass deviation was automatically determined from the first data file for thermos files and set to a mass deviation of 15 ppm for files acquired by bruker instruments. Further calculations were performed

within R (version 4.2.2) using the following libraries: diann, tidyverse, data.table, samr, vsn, and ggplot2, gprofiler, missForest. Data was further processed using an in house modified R-script based on the version by V. Demichev (Github page, cit MaxLFQ). Data input was filtered for unique peptides, $q$-Value < 0.01, Lib.Q.Value < 0.01, PG.Q.Value < 0.01, Global.Q.Value < 0.01, Quantity.Quality > 0.7, Fragment.count >= 4. Fiber with less than 500 identified proteins were excluded. Protein intensities of each fiber was normalized to the total fiber intensity. Fibers were characterized into Fiber type I (>70% MYH7), Fiber type IIa (>70% MYH2), Fiber type IIx (>70% MYH1), and mixed Fiber (<70% MYH1, <70% MYH2, <70% MYH4, <70% MYH7). Data completeness of 70% was calculated on each group. One group is specified by season and fiber type. Missing values were imputed by random forest algorithm for groups with 70% data completeness, with more than 30% missing values random forest algorithm was downshifted of 0.3, width 1.5. Further analysis was performed in perseus (V 1.6.5.0) and InstantClue (V 0.10.10.20211105)[58]. ANOVA analysis and Welch's T-test analysis was performed with an FDR with less than 0.05 and 500 randomizations, Quality control of iRT peptides were performed in Skyline-Daily (V 22.21.391).

### Electron microscopy

Small bundles (about 15–20 myofibers) were dissected from each skinned bear biopsy and pinned, slightly stretched, over a silicone support (SYLGARD 184 Silicone; GMID 01673921). The pinned bundles were then washed with relaxing buffer (potassium propionate 100 mM, MOPS 50 mM, EGTA 12 mM, magnesium acetate 6 mM, potassium dihydrogen phosphate 6 mM, ATP 4 mM, DTT 2 mM, protease inhibitor, pH 7.4) to remove the storage solution while avoiding spontaneous fiber contraction. Still stretched, the samples were fixed overnight at 4 °C with 2.5% glutaraldehyde (EMS; Cat. N°16220) in 0.1 M sodium cacodylate buffer (pH 7.4). Subsequently the samples ware postfixed with 1% in 0.1 M sodium cacodylate buffer for 1 h at 4 °C. After three water washes, samples were dehydrated in a graded ethanol series and embedded in an epoxy resin (Sigma-Aldrich 46345). Ultrathin sections (60–70 nm) were obtained with Leica Ultracut EM UC7 ultramicrotome, counterstained with uranyl acetate and lead citrate and viewed with a Tecnai G2 (FEI) transmission electron microscope operating at 100 kV. Images were captured with a Veleta (Olympus Soft Imaging System) digital camera.

### Immunohistochemistry

Each fiber was dissected from the bear biopsies and pinned over a silicone support (SYLGARD 184 Silicone; GMID 01673921) and washed with relaxing buffer (Potassium propionate 100 mM, MOPS 50 mM, EGTA 12 mM, magnesium acetate 6 mM, potassium dihydrogen phosphate 6 mM, ATP 4 mM, DTT 2 mM, protease inhibitor, pH 7.4) and slightly stretched. The fibers were then fixed in paraformaldehyde 2% for 5 min and permeabilized with PBS Triton X-100 1% for 3 min. The samples were blocked with mouse-on-mouse blocking reagent (Vector laboratory MKB-2213) for 1 h at room temperature. TOM20 primary antibody (Proteintech, Nr 11802-1-AP, lot 00128108) (dilution rate 1:50) was incubated as a cocktail solution of PBS (Sigma-Aldrich, Life science, P4417) 0.5% BSA (Sigma-Aldrich, Life science, A3912) and 2% goat serum (Sigma-Aldrich, Life science, G9023) overnight at 4 °C. Alexa Fluor® 647 conjugated Anti-Rabbit IgG (Alexa Fluor® 647 AffiniPure™ Goat Anti-Rabbit IgG H + L, Jackson ImmunoResearch, 111-605-144, lot 000000142480; 1:100 dilution) secondary antibody was subsequently incubated together with a cocktail solution of Phalloidin (Alexa Fluor 568 phalloidin, Invitrogen, Life Technologies corporation, A1238; lot 2077757, 1:1000 dilution), BSA 0.5% and 4% goat serum for 1 h at 37 °C. After each step the fibers were washed three times in PBS. Lastly, each fiber was carefully placed on microscope slides (series 3 adhesive, Trajan, T7611) and mounted with Elvanol (0.01 g/vol

polyvinyl alcohol, 30% glycerol, PBS). Images were acquired with a confocal scanning laser microscope (Zeiss LSM900 upright confocal).

### Statistics & reproducibility

Statistical analysis has been carried out using GraphPad Prism software (version 10.0.2 for Mac, GraphPad Software, Boston, Massachusetts USA). Individual datasets were tested for outliers using the ROUT method Q = 1%, then tested for normal distribution using D'Agostino & Pearson test before proceeding to parametric or nonparametric unpaired Student $t$-test. Statistical significance is reported for $p$-value < 0.05. Data are presented as means ± standard error of the mean (SEM), and individual data points are shown to visualize distribution. In Figs. 1a, 1b and 2a the histograms on the right report a single value for each bear which was obtained as the mean of the multiple single fiber analysis represented in the histograms on the left. In Figs. 1c and 2b each dot represents a single fiber belonging to all the subjects analyzed within the corresponding season ($n = 2±5$ fibers per subject per season). In Figs. 4b, 5b, 5d and 5e, the bilateral ratio paired Student's $t$-test was used to compare the muscles of bears during summer and winter or boot versus control leg in mice.

### Reporting summary

Further information on research design is available in the Nature Portfolio Reporting Summary linked to this article.

## Data availability

The data generated in this study are provided in the Supplementary Information/Source Data file. The mass spectrometry proteomics data have been deposited to the ProteomeXchange Consortium via the PRIDE partner repository with the dataset identifier PXD050980[57]. Source data are provided with this paper.

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

## Acknowledgements

This work was supported by grants from Association Française contre les Myopathies (AFMTéléthon to B.B., no. 24357), Associazione Italiana per la Ricerca sul Cancro (AIRC)-27007 to B.B. This work is supported by the European Union – NextGenerationEU, by the 2021 STARS Grants@Unipd program and Association Française contre les Myopathies (AFMTéléthon, no. 24328) granted to L. N., This work was supported by the European Union EU-NMJ-Chip (JPND2019-466–146), EU-NMJ-Chip (FKZ:01ED2006), and the Deutsche Forschungsgemeinschaft DFG FOR2722/2 (DFG 384170921) to MK. French Space Agency (CNES, #7906 and #8072) to F.B., and Agence Nationale de la Recherche (ANR-22-CE14-0018) to F.B. and E.L. This work was funded by the European Union via the Horizon 2020 Research and Innovation Program under the Marie Sklodowska-Curie grant agreement no. 886232 to L. M. The authors thank the field capture team of the Scandinavian Brown Bear Research Project (SBBRP). The long-term funding of SBBRP has come primarily from the Swedish Environmental Protection Agency, the Norwegian Environment Agency, the Austrian Science Fund, and the Swedish Association for Hunting and Wildlife Management.

## Author contributions

L.N., B.B., M.K. conceived the project and wrote the manuscript. C.DN., L.S., M.M., L.C., S.S., L.M., E.G., J.K., AL.E., G. GK., M.N. F.B., and E.L. performed experiments and analyzed data. All authors discussed the results and commented on the manuscript.

## Competing interests

The authors declare no competing interests.
