## [Transparent Peer Review file · Nature Communications]

Reduced ATP turnover during hibernation in relaxed skeletal muscle

Corresponding Author: Professor Bert Blaauw

Version 0:

Reviewer comments:

Reviewer #1

(Remarks to the Author)

De Napoli and coworkers investigate the ATP turnover during hibernation in muscle fibers of the brown bear by comparing single muscle fibers obtained during summer and winter from the same bears. By various analyses fibers obtained during hibernation is shown not to be smaller and have nearly maintained force generating capacity, as well as a reduced myosin ATPase turnover. By proteomic analyses remodeling of fibers was observed. Independent of fiber types mitochondria related proteins was altered/decreased in hibernating muscle. One down-regulated protein was MYLK2 – a proposed regulated of myosin ATPase activity. Based on these data the authors hypothesize that the reduction in MYLK2 links to the decreased ATPase turnover and that this lessen energy turnover and thus muscle wasting during hibernation.

This is an interesting study with some potential impact also for human physiology/pathophysiology.

Is it possible to provide causality between the changes in MYLK2 (expression/phosphorylation) and the myosin ATPase activity? This would make the mechanistic insight much stronger. Also, can these findings be reproduced in other muscle of other hibernating animals or in muscle of humans following prolong/extreme fasting or physical in-activity. This would provide more evidence and support the use of the term “evolutionary adaptations”

Given the mitochondria related phenotype – why did the authors not perform measures of muscle endurance rather than force capacity?

L125 – the word “leading” suggests causality – this is not provide in the current study.

L547 - There seem to be something wrong with this sentence

L557 – “the major finding...is that relaxed myosin consumes less ATP during hibernation” – I do not agree that the study provides direct evidence for this conclusion. No ATP turnover is measured.

L591 – The causality is not provided

I am concerned to the validity of the statistical analyses performed when multiple fibers obtained from the same muscle biopsy is compared. As an example Fig 2B.

It might be fair to state that mainly the data from two bears is driving the differences in ATPase activity – this should be discussed. In line, is it possible to calculate the estimated decrease in in vivo energy expenditure due to this less myosin ATPase activity – can this give any meaning full insights?

Given that the antibodies used for MLC2 phosphorylation and MYLK2 likely is raised in another species – has the authors been able to perform any kind of validation of these?

Reviewer #2

(Remarks to the Author)

See attached file.

Version 1:

Reviewer comments:

Reviewer #2

(Remarks to the Author)

The authors have revised the manuscript sufficiently in my opinion. I thank the authors for their attention to detail when addressing the comments.

Reviewer #3

(Remarks to the Author)

Evaluation of answers of comment from reviewer #1.

Reviewer #1 complimented the study but also raised fundamental concerns about the paper.

The authors finding that MYLK2 expression in skeletal muscle was elevated during hibernation was an observation from the proteomic analysis and necessitates documentation of causality for the lower ATP turnover in muscles from bears during hibernation compared to the active summer period. The elevated myosin light chain phosphorylation during hibernation is not considered a valid mechanism for the reduced ATP turnover.

The authors do not provide data documenting causality between increased MYLK2 expression and lower ATP turnover during hibernation.

Instead, the authors argue for such a causality based in existing literature. However, the authors seem to be unaware of the paper by Lewis et al. "Remodeling of skeletal muscle myosin metabolic states in hibernating mammals" (Elife., 2024, May 16:13:RP94616. doi: 10.7554/eLife.94616) with preprint November 2023. This study showed preservation of myosin structure in *U. arctos* and *U. americanus* during hibernation, which completely contrasts that the authors of the present study argue for.

Reasons for the contrasting findings need to be addressed.

Version 2:

Reviewer comments:

Reviewer #3

(Remarks to the Author)

The authors provide no new data to support the mechanistic link between the increased expression of MYLK2 and SRX transformation and reduced ATP consumption. Instead, more literature supporting for their interpretation is included.

The paper by Lewis et al. is now briefly discussed. However, the argument that snap freezing being the reason is highly speculative. Snap freezing should prevent ice crystal formation. Indeed, it can be another issue for the myosin DRX and SRX formation, but this should be confirmed at least with a reference.

Below we responded in cursive and bold to the comments raised by the reviewers:

Reviewer #1 (Remarks to the Author):

De Napoli and coworkers investigate the ATP turnover during hibernation in muscle fibers of the brown bear by comparing single muscle fibers obtained during summer and winter from the same bears. By various analyses fibers obtained during hibernation is shown not to be smaller and have nearly maintained force generating capacity, as well as a reduced myosin ATPase turnover. By proteomic analyses remodeling of fibers was observed. Independent of fiber types mitochondria related proteins was altered/decreased in hibernating muscle. One down-regulated protein was MYLK2 – a proposed regulated of myosin ATPase activity. Based on these data the authors hypothesize that the reduction in MYLK2 links to the decreased ATPase turnover and that this lessen energy turnover and thus muscle wasting during hibernation.

This is an interesting study with some potential impact also for human physiology/pathophysiology.

Is it possible to provide causality between the changes in MYLK2 (expression/phosphorylation) and the myosin ATPase activity? This would make the mechanistic insight much stronger. Also, can these findings be reproduced in other muscle of other hibernating animals or in muscle of humans following prolong/extreme fasting or physical in-activity. This would provide more evidence and support the use of the term “evolutionary adaptations”

An increase in resting ATP consumption due to MYLK activity has been reported in cardiac preparations ¹. Support for the importance of the specific phosphorylation site comes from a work in which an amino acid substitution of the serine into an aspartate (i.e. generating a phospho-mimetic mutant) was sufficient to increase ATPase activity of resting myosin ². Myosin RLC phosphorylation has already been demonstrated to induce an SRX inhibition ³, in addition to a structural disordering of the resting thick filament ⁴, therefore strongly supporting that resting myosin has increased ATPase activity when the light chain is phosphorylated. It has also been reported that myosin phosphorylation leads to a decrease in the energy cost of active contraction ⁵, in line with the increased calcium sensitivity induced by pRLC ⁶⁻⁸. A higher calcium sensitivity allows the muscle to produce the same amount of force with a lower cytosolic free calcium, e.g. a lower motor neuron stimulation rate and SERCA activity, these aspects might play a secondary role as muscle activity during hibernation is very low.

The reduction in MYLK2 protein level highlighted in our work is supported by transcriptional data on skeletal muscle of hibernating and active bear, in which the transcript in summer samples is about 2 times that of winter ⁹. Furthermore, hypothermia induced torpor in zebrafish also causes a decrease in skeletal muscle MYLK2 ¹⁰. We added a phrase citing the results of these transcriptomics studies in the ‘results’ section.

As suggested by the reviewer we checked whether MYLK2 levels could be altered in other disuse models. We found that MYLK2 is reduced about 7 folds in tibialis anterior muscle of denervated mice ¹¹ compared to control. Next, we analyzed a disuse model in mice to determine if similar changes can be observed also there. In our lab, we recently developed a skeletal muscle disuse model based on a 3d printed cast that allows unilateral immobilization ¹² and on those samples we measured a significant decrease in MYLK2 protein level after 7 days of immobilization, which is even more evident after 14 days of immobilization (results showed below). These findings suggest that reducing mylk2 levels occurs in multiple models of muscle disuse. We added these results to figure 5D and in the text in the ‘results’ and ‘discussion’ sections.

Given the mitochondria related phenotype – why did the authors not perform measures of muscle endurance rather than force capacity?

Unfortunately, the skinned fiber model does not allow for analyses of endurance capacity. Exogenous addition of ATP and calcium in the medium to induce contraction only allows for analysis of contractile kinetics and force generation capacity.

L125 – the word “leading” suggests causality – this is not provide in the current study.

The sentence has been corrected.

L547 - There seem to be something wrong with this sentence

The sentence has been corrected.

L557 –“ the major finding....is that relaxed myosin consumes less ATP during hibernation” – I do not agree that the study provides direct evidence for this conclusion. No ATP turnover is measured.

The ATP turnover rate has been measured using two technical approaches: one reported in figure 2A regarding resting myosin ATP consumption, while in figure 2B we performed fluorescent ATP analogue to dissect the stability of individual myosin subpopulations (DRX and SRX).

The rate of ATP hydrolysis showed in Figure 2A was measured using the double reaction assay based on NADH oxidation coupled with ATP utilization by myosin. The assay has been applied in many studies on muscle preparations¹³⁻¹⁷ but we adapted it to the screen of mechanically isolated fibers loaded on a 384 wells plate. The assay is performed on a single fiber in relaxing conditions (relax buffer reported in material and methods), NADH, phosphoenolpyruvate (PEP), Lactate dehydrogenase (LDH) and pyruvate kinase (PK). ATP is hydrolyzed by myosin and the resulting ADP is used by PK together with PEP to generate pyruvate, which is then converted into lactate by LDH which also oxidizes NADH to NAD+ in the same reaction (scheme below, enzymes in red). NADH oxidation rate (measured as 340 nm absorbance over time) is coupled with myosin ATP hydrolysis rate.

As a complementary description of ATP turnover, we performed the mantATP chase assay on single fibers and the results are reported in Figure 2B. In the assay a fluorescent analogue of ATP (mantATP) is placed in a relaxing buffer instead of the normal ATP, and the fiber is incubated so that all ATPases are saturated by the fluorescent analogue. The buffer is then rapidly exchanged with a fresh one containing only normal ATP. Hydrolyzed mantADP diffuses out of the fiber according to the stability of the bound/ATPase, and its fluorescent signal decay is measured with a fluorescent microscope. The decay signal is fitted with a multiple exponential decay function to identify myosin populations and time constants^{3,18}. The mantATP (or mantADP) fluorescent emission is about 10 times stronger when bound to myosin than when exposed to the solvent, limiting the background signal of free diffusing nucleotides.

To better highlight this approach the sentence was corrected in “Both an ATPase activity measurement (NADH-oxidation) and a MantATP chasing approach clearly showed that myosin in single muscle fibers taken from hibernating bears consumes less ATP”.

L591 – The causality is not provided

The sentence has been corrected.

I am concerned to the validity of the statistical analyses performed when multiple fibers obtained from the same muscle biopsy is compared. As an example Fig 2B.

As suggested by the reviewer, the data has been reformatted to be consistent with the previous analysis, and for better representing the difference highlighted. As shown in the new graphs reported also here below, the significance is maintained. Unfortunately, one of the technical requirements of the chasing experiment is that to have fibers that are long enough to be mounted on a setup (>1mm), so the experiment could not be carried out in all samples (n=5 active and n=6 hibernation).

The same correction has been introduced also in figures reporting mantATP chasing populations (supplementary Fig 2A) and Ktr results (Figure 1C), so that now each data point represents the average of the results obtained from a single bear.

It might be fair to state that mainly the data from two bears is driving the differences in ATPase activity – this should be discussed.

As reported in the paper, active animals may have very different activity levels compared to the homogeneity of the winter hibernating period. The two values referred by the reviewer are, in fact, particularly high, but they have been measured multiple times in a consistent number of fibers, and they were not excluded by the identify outliers analysis performed (ROOT test, Q=1%) being not incompatible with a normal distribution considering the standard deviation of the dataset. The data obtained by these two bears clearly contribute to the whole data set collected,

but to show that they do not actively drive the reported difference between summer and winter, we excluded those in the graphs reported here below. The significance of the comparison between season is maintained in both separated seasons and the combined data.

In line, is it possible to calculate the estimated decrease in in vivo energy expenditure due to this less myosin ATPase activity – can this give any meaning full insights?

Most of the literature regarding metabolism and energy utilization of skeletal muscle cells is focused on the balance and rearrangements during exercise¹⁹ to supply energy to excitation-contraction coupling and contracting myofilaments. The available resources covering energetic management in resting muscle tissue are very limited, thus, our estimation needs to fill up the blanks with reasonable guesses.

According to our chasing results, the difference in time constant between myosin DRX and SRX leads to a theoretical decrease in ATP consumption which is of about 25% in disordered myosin heads population and 39% due to super relaxed myosin comparing summer and winter samples. Assuming no significant difference in populations entity, the energy saving of the two populations combined can be estimated as a 28% reduction in biopsies of hibernating bears compared to active ones.

*If we multiply the rates obtained by our resting ATPase activity (0.052 ATP/s⁻¹ during active season and 0.032 ATP/s⁻¹ during hibernation) for a whole day we obtain the consumption of 4493 ATP during active season and 2765 ATP during hibernation, values normalized for a single myosin head. The caloric contribution of 1 mole of ATP is estimated to be 7.3 kcal/mol. An efficiency of 0.4 has been reported for skeletal muscle during contraction¹⁵. In these calculations an efficiency of 0.5 has been assumed, since no mechanical force is exerted. It is possible now convert the consumption of ATP in energetic contribution of a single myosin head, being that 2.7E-20 kcal for summer biopsies and 1.68E-20 kcal for hibernating muscle. To estimate the number of myosin head we can apply the in vivo myosin concentration of 150µM²⁰ to the muscle mass estimated for an average bear in our study which is 19.5Kg (40% of the average body weight during active period which states at 48.8 Kg). The total number of myosin heads is estimated to be 1.7E21. Now it is possible to multiply the energetic contribution of a single myosin head for the number of heads, obtaining the overall energy contribution of 48 kcal/day for active season and 29.6 kcal/day during hibernation. For a bear of average weight of those reported in this work (48.8 Kg during active period), the estimated DEE is about 1940 kcal/day in the active season, and it goes down to 485 kcal/day during hibernation. These values were obtained by considering the active oxygen consumption of 0.276 ml g⁻¹ h⁻¹ and an oxygen consumption during hibernation of 0.069 ml g⁻¹ h⁻¹. The conversion was calculated considering the use of 1L of oxygen to obtain 5 kcal. From measurements on human tissues, resting skeletal muscle energy expenditure is estimated to be 13 kcal/kg*day and in our case is applied to a bear muscle mass of about 19.5Kg as stated earlier. Thus, resting muscle energy expenditure contributes to about 254 kcal/day in active season, and 63.5 kcal/day during hibernation (assuming a decrease to 25%²¹. Combining the two results we obtain a contribution of myosin resting state which accounts for 18% of the tissue DEE (48kcal over 254kcal) during summer, and 47% of tissue DEE during winter (30kcal over 63.5kcal). So, based on ATPase results, the energy sparing of hibernating muscle compared to active is estimated to be 38%.*

This rough esteem could not consider the possible interaction of shivering events and active muscle movements which could potentially induce a temporary shift of myosin heads towards a more disordered state, populating DRX in vivo. Moreover, our analysis was performed at a temperature that is below the physiological, and these calculations were done without accounting for differences in body temperature during activity and hibernation. Nevertheless, we found a convergent estimation on the energy saving extent applying both mantATP chasing and

combined ATPase rates, stating respectively 28% and 38%. Despite having profound limitations, this estimation leads to the idea that an energy sparing contribution of myosin SRX exists, and it could be in the range of 30%.

We added this estimation in supplementary figure 6, using a graphical explanation.

Given that the antibodies used for MLC2 phosphorylation and MYLK2 likely is raised in another species – has the authors been able to perform any kind of validation of these?

We have performed western blotting analysis on mouse muscles and clearly show the efficacy of both antibodies. This is in line with the epitope recognition sequence in mouse/human, which is very similar to bear muscle:

1) Epitope recognition sequence of antibody MYLK2 HPA059704 myosin light chain kinase 2 Anti-MYLK2 Antibody

IEFQAVPSEKSEVGOALCLTAREEDCFQILDCCPPPPAPFPHRMVELRTGNVSSEFSMNSKEALGGGKFGAVCTCMEKAT

91,3 % of homology

tr A0A452SE02 A0A452SE02_URSAM sp Q8VCR8 MYLK2_MOUSE	MATENGAVELGIQSPSTDKAPKGPAGEGPPAAGKESGPPDPKKGPRPPDG MTTENGAVELGSQSLSTEQTPKAAAGDGPSEASEKEPSAPATEKDLSPNA *:***** ** *:::**..**:*:*: **..* ::* **..
tr A0A452SE02 A0A452SE02_URSAM sp Q8VCR8 MYLK2_MOUSE	KKEPGTDPGKKEPDPPTLKKDDKAAALKKEDGALAQPSTSSQAPEGEGHS KKDPGAPDPKNPDPSSLKKDPAKAPGPEKKGDPVPASASSQGPSGEGDG **:*:*:* *:::****:**** * . :.:* . .*:***.*.***..
tr A0A452SE02 A0A452SE02_URSAM sp Q8VCR8 MYLK2_MOUSE	GGGPAEGSAGQPAALPQQTATAEASVEKPDAKQEASGSQGPGEPRVHKKK GGGPAEGSEGPPAALPLPTATAEASIQKLDPTQAPSGNQSGEAKAGKKA ***** * ***** *****:.* *..* .**.*.*.*.:. **
tr A0A452SE02 A0A452SE02_URSAM sp Q8VCR8 MYLK2_MOUSE	AEQAGGRRASPFLHSPSCPAVISRPEKLPVENPLNEASELIFEGVPVT AECREAGRRGSPFLHSPSCPAAIISCSEKTLAVKPLSETTDLVFTGVSVT ** : .***.*****:*** ** . :***:***:*** ** **
tr A0A452SE02 A0A452SE02_URSAM sp Q8VCR8 MYLK2_MOUSE	PGPTDPGPAQVAEGGKRIVADSQKEAGEKAAGQAGQAKVQGDTSRG IEFQ PDPQDPGPVKAG--GTNALAEKKKKEEAKEKASQAGQAKVQGDTPQRIGFQ *. * ***. :.. *.. :*:::** .***:*****:***. : * **
tr A0A452SE02 A0A452SE02_URSAM sp Q8VCR8 MYLK2_MOUSE	AVPSEKSEVGOALCLTAREEDCFQILDCCPPPPAPFPHRIVELRPGNVNS AVPSERVEVGOALCLTAREEDCFQILDCCPPPPAPFPHRIVELRTGNVNS *****: *****:*****:*****:*****:*****:*****
tr A0A452SE02 A0A452SE02_URSAM sp Q8VCR8 MYLK2_MOUSE	QFSMNSKDALGGGKFGAVCTCTEKATGLKLAAKVIKKQTPKDKEMVMLEI EFSMNSKEALGGGKFGAVCTCTERATGLKLAAKVIKKQTPKDKEMVLEI :*****:*****:*****:*****:*****:*****
tr A0A452SE02 A0A452SE02_URSAM sp Q8VCR8 MYLK2_MOUSE	EVMNQLNHRNLIQLYAAIETSHEIVLFMEYIEGGELFERIVDEDEDYQLTEV EVMNQLNHRNLIQLYAAIETSHEIILFMEYIEGGELFERIVDEDEDYHLTEV *****:*****:*****:*****:*****:*****
tr A0A452SE02 A0A452SE02_URSAM sp Q8VCR8 MYLK2_MOUSE	DTMVFVRQICDGI LFMHKMRVLHLDLKPENILCVNTTGHMVKIIDFGLAR DTMVFVRQICDGI LFMHKMRVLHLDLKPENILCVNTTGHVVKIIDFGLAR *****:*****:*****:*****:*****:*****
tr A0A452SE02 A0A452SE02_URSAM sp Q8VCR8 MYLK2_MOUSE	RYNPNEKLVNFGTPEFLSPEVVNYDQISDKTDMWSMGVITYMLLSGLSP RYNPNEKLVNFGTPEFLSPEVVNYDQISDKTDMWSLGVITYMLLSGLSP *****:*****:*****:*****:*****:*****
tr A0A452SE02 A0A452SE02_URSAM sp Q8VCR8 MYLK2_MOUSE	FLGDDDTETLNNVLSANWYFDEETFEAVSDEAKDFVSNLIVKDQRARMNA FLGDDDTETLNNVLSANWYFDEETFEAVSDEAKDFVSNLLTKDQSARMSA *****:*****:*****:*****:*****:*****
tr A0A452SE02 A0A452SE02_URSAM sp Q8VCR8 MYLK2_MOUSE	AQCLAHPLNNAEKAKRCNRRLKSQIILLKLYMKRRWKNFIAVSAANR EQCLAHPLNNAEKAKRCNRRLKSQIILLKLYMKRRWKNFIAVSAANR *****:*****:*****:*****:*****:*****
tr A0A452SE02 A0A452SE02_URSAM sp Q8VCR8 MYLK2_MOUSE	FKKISSSGALMALGV FKKISSSGALMALGV

2) Epitope recognition sequence of antibody phosphor RLC Phospho-Myosin Light Chain 2 (Ser19) Antibody #3671 not available:

Human and bear: 99,4% homology

100% Human and mouse

>sp|O14950|ML12B_HUMAN Myosin regulatory light chain 12B OS=Homo sapiens OX=9606
GN=MYL12B PE=1 SV=2
MSSKKAKTKTTKKRPQRATSNVFAMFDQSQIQEFKEAFNMIDQNRDGFIDKEDLHDMLAS
LGKNPTDAYLDAMMNEAPGPINFMTMFLTMFGEKLNGLTDPEDVIRNAFACFDEEATGTIQE
DYLRELLTTMGDRFTDEEVDELYREAPIDKKGNFNIEFTRILKHGAKDKDD

>sp|Q3THE2|ML12B_MOUSE Myosin regulatory light chain 12B OS=Mus musculus
OX=10090 GN=My112b PE=1 SV=2
MSSKKAKTKTTKKRPQRATSNVFAMFDQSQIQEFKEAFNMIDQNRDGFIDKEDLHDMLAS
LGKNPTDAYLDAMMNEAPGPINFMTMFLTMFGEKLNGLTDPEDVIRNAFACFDEEATGTIQE
DYLRELLTTMGDRFTDEEVDELYREAPIDKKGNFNIEFTRILKHGAKDKDD

>tr|A0A452QQX7|A0A452QQX7_URSAM Myosin light chain 12B OS=Ursus americanus
OX=9643 GN=MYL12B PE=4 SV=1
MSSKKAKTKTTKKRPQRATSNVFAMFDQSQIQEFKEAFNMIDQNRDGFIDKEDLHDMLAS
LGKNPTDAYLEAMMNEAPGPINFMTMFLTMFGEKLNGLTDPEDVIRNAFACFDEEATGTIQE
DYLRELLTTMGDRFTDEEVDELYREAPIDKKGNFNIEFTRILKHGAKDKDD

Sequences multi alignment:

```
sp|O14950|ML12B_HUMAN          MSSKKAKTKTTKKRPQRATSNVFAMFDQSQIQEFKEAFNMIDQNRDGFID
sp|Q3THE2|ML12B_MOUSE          MSSKKAKTKTTKKRPQRATSNVFAMFDQSQIQEFKEAFNMIDQNRDGFID
tr|A0A452QQX7|A0A452QQX7_URSAM MSSKKAKTKTTKKRPQRATSNVFAMFDQSQIQEFKEAFNMIDQNRDGFID
*****
```

Reviewer #2 (Remarks to the Author):

Summary:

The manuscript by Napoli, Schmidt et al. presents new information regarding the remodeling of skeletal muscle in hibernating bears. The study is well designed, albeit with a typical caveat that the number of animals is small (9). Nevertheless, multiple measurements for several parameters enhance this reviewer's confidence in the results. Most of the methods are robust, and the results are presented in a clear and informative manner. The conclusions are valid and expand our understanding of the underlying biochemical, biophysical, and anatomical changes occurring in skeletal muscles in bears that hibernate. However, a greater integration of energetics (based on the western results of CI-CV proteins and immunohistochemistry evidence of reduced numbers of mitochondria in hibernation) and expansion to include the results from several earlier muscle studies in bears is requested.

To better show the differences in mitochondrial proteins between summer and winter bears, we made the following scheme which we placed in figure 4D. As can be seen, proteins linked to fat oxidation are reduced while glycolytic proteins are maintained. We added a phrase stating this to the results section.

(green down in winter compared to summer, red contrary and white, unchanged or NF)

Major Comments:

Materials and Methods

L. 131 - The use of young bears (ages 2-3 years) should be expanded to indicate whether they are sexually mature. This is an important consideration as the hormonal milieu, especially testosterone, could impact skeletal muscle dynamics.

The reviewer pointed out some important points regarding age of the subject and their reproductive stage. Brown bears are usually sexually mature at the age of three to four at minimum, the ones considered in this study are, thus, not susceptible to breeding. The bears included in this study are followed since birth to be completely sure that they are not invested in reproductive activities and, thus, not to interfere with possible ongoing reproduction processes. Moreover, the fact that these animals are juveniles limits the danger during the catching for the scientific and veterinary equipe, and it allows to limit the quantity of anesthetics.

It has been shown that testosterone level in 2 to 3 years old bears, as of those included in our study, is two times higher in males in the active period²². In the hibernation period, test levels tend to increase in males but not significantly whereas the levels drop significantly in females.

Serum samples from the same bear group has been assessed by our collaborators recently (unpublished results) and, contrary to the previously cited paper, test levels were higher in females vs males during the active period. The summer vs winter difference in males was not reported, while that in females was confirmed.

Our cohort of study is composed of only 2 males, while we analyzed 7 females, possibly leading to a bias toward females. This could result in an overall drop in testosterone in winter for most of the animals included in the paper. Since testosterone is expected to increase muscle mass, it is not expected to be of major importance in our bears. Of course, this does not preclude a possible importance in adult bears.

We added a phrase to the results section to point out the sexual immaturity of the bears examined.

L. 135 - The authors should also specify if the captures during the active season were performed via helicopter darting or trapping.

As suggested by the reviewer, details on the capture have been added to the methods section and can be found in²³

L. 146 - Please provide a reference for the skinning solution used.

The skinning solution used in this work is an adaptation of that reported in ^{14,24}. We added these references to the methods section

L. 151-170 – Please provide information regarding the controls used for western blot experiments. For example, omission of primary antibody, preabsorption, etc. Also, please indicate the species for primary and secondary antibodies. A reference would also be helpful. Please provide information on the protein stain used in Figure 5B (lower panel, gel stain). Please indicate the dilution used of secondary antibodies.

We added the data in the table below in the materials and methods section describing the western blotting. In figure 5B, like in all other blots we normalized for protein loading as determined by ponceau staining.

Primary antibody			Secondary antibody	
Reference antibody	Dilution	Host	Dilution	Specie
Phospho-Myosin Light Chain 2 (Ser19)(CST#3671, Cell Signaling Technology, Milan, Italy)	1:1000 5% BSA	Rabbit	1:4000 5% milk	Anti-rabbit
MYLK2 (HPA059704, Atlas antibodies, Rome, Italy)	1:1000 5% BSA	Rabbit	1:4000 5% milk	Anti-rabbit
OXPPOS : Abcam mouse cocktail anti-OXPPOS [oxidative phosphorylation] (#ab110413 Abcam, Milan, Italy)	1:2000 5% BSA	Mouse	1:5000 5% BSA	Anti-mouse

L. 211-227 – The preparation appears not to consider changes in protein content with season as observed previously (Miyazaki et al., 2022; Hershey et al., 2008; Tinker et al., 1998). Thus, have the authors attempted to normalize the ATPase activity data? This is a similar concern for the proteomic study (L. 247-254).

ATPase activity data are normalized against the specific myosin content. A single fiber is pipetted in a well of a 384 well plate, and the double reaction mix is added (pyruvate kinase, lactate dehydrogenase, NADH and PEP). The release of ADP due to myosin activity is measured as NADH oxidation for about 40 minutes at 27°C. At the end of the measurement, a concentrated KCl solution is added to each well to reach 0.4M final concentration and induce myosin solubilization from the fiber. The Pierce™ 660nm Protein Assay Reagent (Thermofisher nr. 22660) is then added to each well to estimate the amount of extracted myosin.

Regarding the proteomic analysis, the concern raised by the reviewer is relevant and we faced it applying a specific normalization strategy.

While other proteomics studies on single muscle fibers normalize protein intensities using "housekeeping" proteins like ACTA1 ²⁵, we normalize each protein intensity to the summarized intensity of all quantified proteins in one fiber. A previous study has already shown that this normalization strategy works with different input materials, even with significant changes in protein content and intensities ²⁶. With this centering normalization strategy, we can effectively address changes in protein intensities while still being aware of individual protein content changes. The normalization strategy is explained in the methods part for Single fiber proteomics.

L. 235 – Please provide a reference for the temperature chosen (17.5C) to perform tension measurements. Was this based on Q10 effects or empirical evidence?

The reviewer addressed an important point about temperature. First, we would like to apologize to the reviewers since the temperature reported in the manuscript does not correspond to that of the experiments, which was 21°C and it has now been corrected in the methods.

The activation of skinned fibers is prone to rundown effects mostly caused by the uneven calcium diffusion, leading to irregular myofibril activation. The shear stress to which the fiber is subjected due to this unsynchronized activation leads to fiber damage, as evidenced by a drop in force produced, with reduced reliability and reproducibility of the measurements. To overcome this issue, the temperature-jump (T-jump) technique has been introduced which limits rundown effect. The T-jump consists in the activation of the fiber in very low temperature, transferring it to high temperature only when calcium is already available. The result is a strong reduction of the shear stress between myofibrils due to slow calcium diffusion, which would need to flow from outside the fiber^{27,28}, and not be instantaneously available.

Force developed by mammalian skinned fibers is affected by temperature following a temperature coefficient Q10 ^{29,30}, which diminishes above 20°C. Maximal tension measurements on skinned fibers is measured as a steady state in which all myosin heads are recruited, completely eliminating myosin SRX when force is higher than 50%

of the maximal tension³¹. In this regard, it is possible to assume that our tension measurements performed at 21°C are representative of a maximal fiber contractile performance.

In our work, the temperature was chosen to be 21°C to find a balance between the reliability of the tension measurements and the maintenance of myosin structural arrangement, limiting fiber damage. In fact, it has been reported that myosin orientation in resting mammalian fibers is susceptible to low temperature³² in which it assumes a more perpendicular/rigor orientation. The force redevelopment obtained by the fast shortening of the fiber is the result of the combination of different effects, including that of SRX/DRX equilibrium³². Thus, to maintain the difference in biochemical states highlighted by the chasing experiment in active and hibernating fibers, the temperature needed to be high enough that the myosin orientation would not interfere with the biochemical equilibrium³³.

In brief, we chose a temperature of 21°C in our experiments to meet the requirement of a preserved myosin DRX/SRX balance between summer and winter samples (best at physiological temperatures), with that of a reliable model not affected by sample rundown (best at low temperature).

L. 292-306 – The preparation of EM tissue would have benefited from a freshly prepared sample process in the same way. The images provided are not very convincing and are somewhat difficult to interpret (see below). How many bears were these taken from? How many from each season?

We have decided to move the EM images to the supplementary figures just to allow the reader to have an idea of the ultrastructure of the main experimental model used in the manuscript (i.e. the skinned muscle fiber). We removed all qualitative/quantitative observations about mitochondria as they are indeed difficult to quantify.

L. 320 – Please indicate the dilution of secondary antibody used. Results

We added this to the methods section.

L. 344 – Would suggest rewording the heading. “Mild” is used without any context and thus is not informative.

We removed this from the title of the chapter.

L. 382 – Please provide a definition for “Ktr”.

We added the following text to the methods section:

Ktr is described as the rate of force redevelopment following a rapid shortening of the fiber after reaching maximal isometric tension. The rapid shortening allows the tension to be redeveloped from zero in a saturating calcium condition so that the kinetic is not affected by Ca²⁺ diffusion, the thin filament is activated, and the force increases proportionally to the thick filament activation and myosin recruitment. The experimental traces are fitted using a single exponential function as reported extensively in the literature³⁴.

L. 389-410 – As indicate above, these results would benefit from a normalization either to total mass or total protein content to ensure that the results do not reflect other changes.

This is an important issue indeed, and as mentioned in the methods sections, we have been very careful to normalize all our measurements for total protein/myosin content.

For force: Force is normalized to cross sectional area using the measured diameter and assuming circular geometry.

For ATPase activity: A concentrated KCl solution was added to each well at the end of the assay to reach the final concentration of 0.4M and extract myosin from the bear fiber, the protein amount was measured using Pierce™ 660nm Protein Assay Reagent (ThermoFisher nr. 22660) for normalization purposes.

For single fiber proteomics: Protein intensities of each fiber was normalized to the total fiber intensity.

For western blotting: Signals were quantified using the ImageJ software 1.53f51³⁵ and normalized against the total amount of proteins determined by Ponceau signals to correct for uneven loading.

L. 468-481 – Have the authors considered quantifying what appears to be a huge difference in mitochondria numbers between seasons (Fig. 4B). If this is not representative, please consider a different image.

The proteomic analysis revealed a significant change in protein associated with mitochondria functionality and structure (Figure 3E) that, in the case of the respiratory complexed does not reach significance if considered individually. The overall decrease in mitochondria functionality agrees with previous publications³⁶ and it was confirmed by our western blot analysis of the respiratory complexes (Figure 4A). We believe that proteomic and western blotting are more representative for quantification purposes, as they analyze many fibers and a larger amount of tissue.

Regarding immunofluorescent analysis, here below, we reported three individual bears stained in winter and summer for TOM20 at 40X and 63X magnification (numbers are indicative of the bear subject and year of sample collection). Despite not being quantitative as intended, in these pictures the mitochondrial density and organization seems to be consistently better in skinned fibers from active bears compared to those during hibernation, also considering they are collected from the same animal.

We think that the TOM20 fluorescent staining offers a useful visual input over the quality of the samples, and it is not meant to be quantitative since the two approaches listed earlier (proteomic analysis and western blotting) are more suitable techniques for quantification purposes.

Additionally, this reviewer is not convinced of the value added by including the EM study as this does not have freshly prepared tissue for comparison. For example, were the number of mitochondria counted and compared to the IHC results? This is actually somewhat surprising given that it would have been available at the time of collection. Suggest removing EM results.

As mentioned previously, we will remove the EM results as they cannot be quantified properly.

Discussion

L. 506-510 - The discussion would benefit from the addition of comparative numbers revealing the extent of muscle loss in other species, with aging, fasting, inactivity, etc. for comparison.

As suggested by the reviewer, now the manuscript include some original data on MYLK2 abundance during disuse induced by a 3d printed cast applied for 1 and 2 weeks in mouse. This model induced a muscle loss of 25% over the course of 2 weeks of inactivity¹². Moreover, the comparative information written here below was added to the initial part of the discussion.

As shown in this manuscript and by others, during hibernation very little if any muscle mass is lost, while muscle strength decreases only about 29% in 110 days³⁷. As a comparison, a 90-days immobilization in human led to a decrease in force of 54%³⁸. In human, the effect of bed-rest and immobilization periods of 4-28 days were reviewed in^{39,40} showing a muscle loss in the range of 0.2-2.3 %/day, always exceeded by a force loss in the range of 1.1-3.5 %/day.

As a comparison, in a disuse model in rodents caused by 3d printed cast, the gastrocnemius muscle exhibits a 25% in weight loss and a 40% drop in muscle force over the course of 2 weeks¹².

Throughout - Other than MYLK2, there is a surprising paucity of information (one sentence, L518-520) in the discussion related to the proteomics results. Further, there is little attempt to integrate this information with earlier studies or integration with the biochemical and biophysical results. This integration is required since it represents an important component. For example, were changes in MAFBX (FBX032), MURF1 (TRIM63), or PGC1a observed? These have been proposed previously as mechanisms to reduce the amount of atrophy (Lin et al., 2012).

We added a new paragraph to the discussion in which we briefly discuss the proteome changes (either up or down) in hibernating muscles, and how these can contribute to some of the changes in muscle physiology we measured.

It was shown previously that the 'classical' atrogenes are not altered in hibernating snap frozen muscles⁹. This lack of atrogene induction is very likely due to the transient nature of this induction. The group of Alfred Goldberg showed how these genes increase their expression rapidly after an atrophic stimulus, and then rapidly

decline in the order of days/weeks depending on the atrophic stimulus⁴¹. Therefore, it is not surprising that there is no change in these atrogenes in hibernating muscles.

L. 575-588 – The discussion of MYLK2 and P-MLC2 results overlooks a potential explanation for the maintenance of ATP levels in winter muscle, namely, that the efficiency of phosphorylation (or dephosphorylation) is altered. Specifically, it appears that the ratio of MYLK2 to PMLC2 is much higher in winter than in summer (Fig. 5C). If this is indeed the case (I'm basing this off of estimates from the average), the potential changes phosphatases in addition to kinases could be important players. A search through the proteomics data might provide important clues to the accuracy of this statement.

We went through the proteomics data, but we could not identify any changes in phosphatase subunits. The reviewer is of course right that this could be part of the explanation, so mention this possibility in the discussion section.

Minor Comments:

L. 167 – with appropriate

The sentence has been corrected.

L. 179 – Samples for protein electrophoresis “were”

The sentence has been corrected.

L. 253-254 – The Hughes reference is incorrectly formatted L. 259 – Define “FA”

The sentence has been corrected.

L. 313 – fixated should be “fixed”

The sentence has been corrected.

L. 382 – please define “Ktr”

We defined this in the methods section now.

L. 445 – mouse is indicated, but no animal protocol is identified for this tissue collection. Under what conditions was this mouse held, strain, etc.?

We added the following phrase to the methods section:

Mouse tissue was obtained from 3-month-old male black 6 mice. Experimental protocols were reviewed and approved by the Ethical Committee, University of Padova (No.225/2022 PR).

L. 571 – Use of “comforted” should be “supported”?

The sentence has been corrected.

L. 654 – If only 9 bears were used in the study, where does the n=11 come from. Presumably this is from counting the two bears that were resampled in year two?

This is because we had one more couple for frozen tissue than for single fibers

L. 661 – Please provide units for the Y-axis of Fig. 2B. Were these normalized data?

The values of the fluorescence decay in figure 2B are normalized on that of the first point of the chasing which has the stronger intensity. We corrected that axis title by changing to “normalized fluorescence intensity”.

This data was normalized to the first value, therefore the Y-axis is normalized data.

L. 718 – From the Venn diagrams in Suppl. Fig. 3D it appears that only a single protein was present in both summer1 and summer2 muscles. This seems unusual. A similar scenario exists for MYH2 winter1 and winter2. Unless I'm mis-reading these, this suggests poor reproducibility. Please clarify.

The parts of the VENN diagram showing a 1 means that there is one protein which is only present in that condition and not in any other. As can be seen by the high number in the center, most proteins are found in all conditions.

1. Muthu, P., Kazmierczak, K., Jones, M., and Szczesna-Cordary, D. (2012). The effect of myosin RLC phosphorylation in normal and cardiomyopathic mouse hearts. *J Cell Mol Med* 16, 911-919. 10.1111/j.1582-4934.2011.01371.x.
2. Yadav, S., Kazmierczak, K., Liang, J., Sitbon, Y.H., and Szczesna-Cordary, D. (2019). Phosphomimetic-mediated in vitro rescue of hypertrophic cardiomyopathy linked to R58Q mutation in myosin regulatory light chain. *The FEBS journal* 286, 151-168. 10.1111/febs.14702.
3. Stewart, M.A., Franks-Skiba, K., Chen, S., and Cooke, R. (2010). Myosin ATP turnover rate is a mechanism involved in thermogenesis in resting skeletal muscle fibers. *Proc Natl Acad Sci U S A* 107, 430-435. 10.1073/pnas.0909468107.
4. Levine, R.J., Kensler, R.W., Yang, Z., Stull, J.T., and Sweeney, H.L. (1996). Myosin light chain phosphorylation affects the structure of rabbit skeletal muscle thick filaments. *Biophysical journal* 71, 898-907. 10.1016/S0006-3495(96)79293-7.
5. Cooke, R., Franks, K., and Stull, J.T. (1982). Myosin phosphorylation regulates the ATPase activity of permeable skeletal muscle fibers. *FEBS letters* 144, 33-37. 10.1016/0014-5793(82)80563-2.
6. Sweeney, H.L., and Stull, J.T. (1990). Alteration of cross-bridge kinetics by myosin light chain phosphorylation in rabbit skeletal muscle: implications for regulation of actin-myosin interaction. *Proc Natl Acad Sci U S A* 87, 414-418. 10.1073/pnas.87.1.414.
7. Metzger, J.M., Greaser, M.L., and Moss, R.L. (1989). Variations in cross-bridge attachment rate and tension with phosphorylation of myosin in mammalian skinned skeletal muscle fibers. Implications for twitch potentiation in intact muscle. *J Gen Physiol* 93, 855-883. 10.1085/jgp.93.5.855.
8. Szczesna, D., Zhao, J., Jones, M., Zhi, G., Stull, J., and Potter, J.D. (2002). Phosphorylation of the regulatory light chains of myosin affects Ca²⁺ sensitivity of skeletal muscle contraction. *Journal of applied physiology* 92, 1661-1670. 10.1152/jappphysiol.00858.2001.
9. Cussonneau, L., Boyer, C., Brun, C., Deval, C., Loizon, E., Meugnier, E., Gueret, E., Dubois, E., Taillandier, D., Polge, C., et al. (2021). Concurrent BMP Signaling Maintenance and TGF-beta Signaling Inhibition Is a Hallmark of Natural Resistance to Muscle Atrophy in the Hibernating Bear. *Cells* 10. 10.3390/cells10081873.
10. Cahill, T., Chan, S., Overton, I.M., and Hardiman, G. (2023). Transcriptome Profiling Reveals Enhanced Mitochondrial Activity as a Cold Adaptive Strategy to Hypothermia in Zebrafish Muscle. *Cells* 12. 10.3390/cells12101366.
11. Henze, H., Huttner, S.S., Koch, P., Schuler, S.C., Groth, M., von Eyss, B., and von Maltzahn, J. (2024). Denervation alters the secretome of myofibers and thereby affects muscle stem cell lineage progression and functionality. *NPJ Regen Med* 9, 10. 10.1038/s41536-024-00353-3.
12. Masiero, G., Ferrarese, G., Perazzolo, E., Baraldo, M., Nogara, L., and Tezze, C. (2024). Custom-made 3D-printed boot as a model of disuse-induced atrophy in murine skeletal muscle. *PLoS One* 19, e0304380. 10.1371/journal.pone.0304380.

13. Griffiths, P.J., Guth, K., Kuhn, H.J., and Ruegg, J.C. (1980). ATPase activity in rapidly activated skinned muscle fibres. *Pflugers Arch* 387, 167-173. 10.1007/BF00584268.
14. Stienen, G.J., Kiers, J.L., Bottinelli, R., and Reggiani, C. (1996). Myofibrillar ATPase activity in skinned human skeletal muscle fibres: fibre type and temperature dependence. *J Physiol* 493 (Pt 2), 299-307. 10.1113/jphysiol.1996.sp021384.
15. He, Z.H., Bottinelli, R., Pellegrino, M.A., Ferenczi, M.A., and Reggiani, C. (2000). ATP consumption and efficiency of human single muscle fibers with different myosin isoform composition. *Biophysical journal* 79, 945-961. 10.1016/S0006-3495(00)76349-1.
16. Parijat, P., Attili, S., Hoare, Z., Shattock, M., Kenyon, V., and Kampourakis, T. (2023). Discovery of a novel cardiac-specific myosin modulator using artificial intelligence-based virtual screening. *Nat Commun* 14, 7692. 10.1038/s41467-023-43538-y.
17. Gyimesi, M., Horvath, A.I., Turos, D., Suthar, S.K., Penzes, M., Kurdi, C., Canon, L., Kikuti, C., Ruppel, K.M., Trivedi, D.V., et al. (2020). Single Residue Variation in Skeletal Muscle Myosin Enables Direct and Selective Drug Targeting for Spasticity and Muscle Stiffness. *Cell* 183, 335-346 e313. 10.1016/j.cell.2020.08.050.
18. Myburgh, K.H., Franks-Skiba, K., and Cooke, R. (1995). Nucleotide turnover rate measured in fully relaxed rabbit skeletal muscle myofibrils. *J Gen Physiol* 106, 957-973. 10.1085/jgp.106.5.957.
19. Hargreaves, M., and Spriet, L.L. (2020). Skeletal muscle energy metabolism during exercise. *Nat Metab* 2, 817-828. 10.1038/s42255-020-0251-4.
20. Ferenczi, M.A., Homsher, E., and Trentham, D.R. (1984). The kinetics of magnesium adenosine triphosphate cleavage in skinned muscle fibres of the rabbit. *J Physiol* 352, 575-599. 10.1113/jphysiol.1984.sp015311.
21. Toien, O., Blake, J., Edgar, D.M., Grahn, D.A., Heller, H.C., and Barnes, B.M. (2011). Hibernation in black bears: independence of metabolic suppression from body temperature. *Science* 331, 906-909. 10.1126/science.1199435.
22. Frobert, A.M., Toews, J.N.C., Nielsen, C.G., Brohus, M., Kindberg, J., Jessen, N., Frobert, O., Hammond, G.L., and Overgaard, M.T. (2022). Differential Changes in Circulating Steroid Hormones in Hibernating Brown Bears: Preliminary Conclusions and Caveats. *Physiol Biochem Zool* 95, 365-378. 10.1086/721154.
23. Stenvinkel, P., Frobert, O., Anderstam, B., Palm, F., Eriksson, M., Bragfors-Helin, A.C., Qureshi, A.R., Larsson, T., Friebe, A., Zedrosser, A., et al. (2013). Metabolic changes in summer active and anuric hibernating free-ranging brown bears (*Ursus arctos*). *PLoS One* 8, e72934. 10.1371/journal.pone.0072934.
24. Germani, S., Van Ho, A.T., Cherubini, A., Varone, E., Chernorudskiy, A., Renna, G.M., Fumagalli, S., Gobbi, M., Lucchetti, J., Bolis, M., et al. (2024). SEPN1-related myopathy depends on the oxidoreductase ERO1A and is druggable with the chemical chaperone TUDCA. *Cell Rep Med* 5, 101439. 10.1016/j.xcrm.2024.101439.
25. Murgia, M., Nagaraj, N., Deshmukh, A.S., Zeiler, M., Cancellara, P., Moretti, I., Reggiani, C., Schiaffino, S., and Mann, M. (2015). Single muscle fiber proteomics reveals unexpected mitochondrial specialization. *EMBO reports* 16, 387-395. 10.15252/embr.201439757.
26. Schmidt, L., Saynisch, M., Hoegsbjerg, C., Schmidt, A., Mackey, A., Lackmann, J.W., Muller, S., Koch, M., Brachvogel, B., Kjaer, M., et al. (2024). Spatial proteomics of skeletal muscle using thin cryosections reveals metabolic adaptation at the muscle-tendon transition zone. *Cell reports* 43, 114374. 10.1016/j.celrep.2024.114374.

27. Davis, J.S., and Harrington, W.F. (1987). Laser temperature-jump apparatus for the study of force changes in fibers. *Anal Biochem* 161, 543-549. 10.1016/0003-2697(87)90487-8.
28. Goldman, Y.E., McCray, J.A., and Ranatunga, K.W. (1987). Transient tension changes initiated by laser temperature jumps in rabbit psoas muscle fibres. *J Physiol* 392, 71-95. 10.1113/jphysiol.1987.sp016770.
29. Bottinelli, R., Canepari, M., Pellegrino, M.A., and Reggiani, C. (1996). Force-velocity properties of human skeletal muscle fibres: myosin heavy chain isoform and temperature dependence. *J Physiol* 495 (Pt 2), 573-586. 10.1113/jphysiol.1996.sp021617.
30. Coupland, M.E., and Ranatunga, K.W. (2003). Force generation induced by rapid temperature jumps in intact mammalian (rat) skeletal muscle fibres. *J Physiol* 548, 439-449. 10.1113/jphysiol.2002.037143.
31. Linari, M., Brunello, E., Reconditi, M., Fusi, L., Caremani, M., Narayanan, T., Piazzesi, G., Lombardi, V., and Irving, M. (2015). Force generation by skeletal muscle is controlled by mechanosensing in myosin filaments. *Nature* 528, 276-279. 10.1038/nature15727.
32. Fusi, L., Huang, Z., and Irving, M. (2015). The Conformation of Myosin Heads in Relaxed Skeletal Muscle: Implications for Myosin-Based Regulation. *Biophysical journal* 109, 783-792. 10.1016/j.bpj.2015.06.038.
33. Xu, S., Offer, G., Gu, J., White, H.D., and Yu, L.C. (2003). Temperature and ligand dependence of conformation and helical order in myosin filaments. *Biochemistry* 42, 390-401. 10.1021/bi026085t.
34. Brenner, B., and Eisenberg, E. (1986). Rate of force generation in muscle: correlation with actomyosin ATPase activity in solution. *Proc Natl Acad Sci U S A* 83, 3542-3546. 10.1073/pnas.83.10.3542.
35. Schneider, C.A., Rasband, W.S., and Eliceiri, K.W. (2012). NIH Image to ImageJ: 25 years of image analysis. *Nature methods* 9, 671-675. 10.1038/nmeth.2089.
36. Chazarin, B., Storey, K.B., Ziemianin, A., Chanon, S., Plumel, M., Chery, I., Durand, C., Evans, A.L., Arnemo, J.M., Zedrosser, A., et al. (2019). Metabolic reprogramming involving glycolysis in the hibernating brown bear skeletal muscle. *Front Zool* 16, 12. 10.1186/s12983-019-0312-2.
37. Lohuis, T.D., Harlow, H.J., and Beck, T.D. (2007). Hibernating black bears (*Ursus americanus*) experience skeletal muscle protein balance during winter anorexia. *Comp Biochem Physiol B Biochem Mol Biol* 147, 20-28. 10.1016/j.cbpb.2006.12.020.
38. Alkner, B.A., and Tesch, P.A. (2004). Efficacy of a gravity-independent resistance exercise device as a countermeasure to muscle atrophy during 29-day bed rest. *Acta Physiol Scand* 181, 345-357. 10.1111/j.1365-201X.2004.01293.x.
39. Wall, B.T., Dirks, M.L., and van Loon, L.J. (2013). Skeletal muscle atrophy during short-term disuse: implications for age-related sarcopenia. *Ageing Res Rev* 12, 898-906. 10.1016/j.arr.2013.07.003.
40. Sarto, F., Bottinelli, R., Franchi, M.V., Porcelli, S., Simunic, B., Pisot, R., and Narici, M.V. (2023). Pathophysiological mechanisms of reduced physical activity: Insights from the human step reduction model and animal analogues. *Acta Physiol (Oxf)* 238, e13986. 10.1111/apha.13986.
41. Lecker, S.H., Jagoe, R.T., Gilbert, A., Gomes, M., Baracos, V., Bailey, J., Price, S.R., Mitch, W.E., and Goldberg, A.L. (2004). Multiple types of skeletal muscle atrophy

involve a common program of changes in gene expression. *FASEB J* 18, 39-51.
10.1096/fj.03-0610com

18/1/39 [pii].

Dear Editor, we thank you for the comments made. Below we responded in cursive and bold to the comments raised by the reviewers:

Reviewer #2 (Remarks to the Author):

The authors have revised the manuscript sufficiently in my opinion. I thank the authors for their attention to detail when addressing the comments.

Reviewer #3 (Remarks to the Author):

Evaluation of answers of comment from reviewer #1.

Reviewer #1 complimented the study but also raised fundamental concerns about the paper.

The authors finding that MYLK2 expression in skeletal muscle was elevated during hibernation was an observation from the proteomic analysis and necessitates documentation of causality for the lower ATP turnover in muscles from bears during hibernation compared to the active summer period. The elevated myosin light chain phosphorylation during hibernation is not considered a valid mechanism for the reduced ATP turnover.

The authors do not provide data documenting causality between increased MYLK2 expression and lower ATP turnover during hibernation.

Instead, the authors argue for such a causality based in existing literature. However, the authors seem to be unaware of the paper by Lewis et al. "Remodeling of skeletal muscle myosin metabolic states in hibernating mammals" (Elife., 2024, May 16:13:RP94616. doi: 10.7554/eLife.94616) with preprint November 2023. This study showed preservation of myosin structure in *U. arctos* and *U. americanus* during hibernation, which completely contrasts that the authors of the present study argue for.

Reasons for the contrasting findings need to be addressed.

The result of an enrichment analysis were reported in Figure 5A. That analysis is based on the list of proteins that showed a significant difference in abundance in the single fiber proteomics analysis. The most significant entry obtained from the enrichment is related to the activity of Myosin Light Chain Kinase 2, which we showed to be reduced in winter sample, both in its total amount and its main target, Myosin Regulatory Light Chain (RLC). Figure 5B showed a western blot analysis of both total MYLK2 and pRLC (Ser19), while quantification is reported in Figure 5C.

The effect of myosin regulatory light chain phosphorylation on thick filament arrangement, post-tetanic potentiation and increased calcium sensitivity is well documented in the literature¹⁻⁵. Regarding ATP consumption, we cited in our work a paper reporting an increased ATP consumption due to RLC phosphorylation in cardiac myofibrils⁶. Despite not being extensively documented for skeletal muscle tissue, it is possible to speculate a similar effect of increased ATP consumption, thus nucleotide turnover, as reported in the original paper describing myosin SRX⁷. We believe the link between phosphorylation of the regulatory light chain and increased ATPase activity is well supported by the reported literature. Furthermore, we now included the reference to a compelling review of Nag and

colleagues, in which the link between SRX and RLC phosphorylation is described in great detail⁸.

The work by Lewis and colleagues only assessed structural changes in sample of the small hibernator *Ictidomys tridecemlineatus* (which is accompanied by a very big drop in temperature, something not observed in bear samples) by X-ray diffraction pattern, while performing mantATP chasing on bear samples. The mantATP chasing is a biochemical assay which measure nucleotide turnover, but does not offer any insight into the structural arrangement of myosin heads. In addition, a recent work supports the idea that closed myosin structure and slow-ATPase biochemical state are not unambiguously associated, quote “These results suggest that the structurally defined interacting-heads motif (IHM) is sufficient but not necessary for the low ATPase activity of the SRX.”⁹.

Another important difference between our work and that of Lewis and colleagues is the fact that our samples went from fresh to skinned without experiencing snap freezing. We believe that the freezing/thawing process could cause ice crystals formation, potentially altering the ordered pattern of thick filaments which has an important role in SRX cooperative nature. We strongly feel that this is an important reason for the differences observed in the chasing experiments. Indeed, they observe a very strong variability in their data, which we feel could be due to this freeze/thaw cycle. Lastly, compared to Lewis and colleagues, our data are supported by a higher number of biopsies, collected in two subsequent seasons which could drive the discrepancy between the obtained outcomes. We added this explanation to the discussion, as indeed it is relevant to try and understand the reasons behind the differences observed in the two studies.

References

1. Persechini, A., Stull, J. T. & Cooke, R. The effect of myosin phosphorylation on the contractile properties of skinned rabbit skeletal muscle fibers. *J Biol Chem* 260, 7951–7954 (1985).
2. Szczesna, D. et al. Phosphorylation of the regulatory light chains of myosin affects Ca²⁺ sensitivity of skeletal muscle contraction. *Journal of Applied Physiology* 92, 1661–1670 (2002).
3. Zhi, G. et al. Myosin light chain kinase and myosin phosphorylation effect frequency-dependent potentiation of skeletal muscle contraction. *Proc Natl Acad Sci U S A* 102, 17519–17524 (2005).
4. Kampourakis, T. & Irving, M. Phosphorylation of myosin regulatory light chain controls myosin head conformation in cardiac muscle. *J Mol Cell Cardiol* 85, 199–206 (2015).
5. Yamaguchi, M. et al. X-ray diffraction analysis of the effects of myosin regulatory light chain phosphorylation and butanedione monoxime on skinned skeletal muscle fibers. *Am J Physiol Cell Physiol* 310, C692–C700 (2016).
6. Muthu, P., Kazmierczak, K., Jones, M. & Szczesna-Cordary, D. The effect of myosin RLC phosphorylation in normal and cardiomyopathic mouse hearts. *J Cell Mol Med* 16, 911–919 (2012).
7. Stewart, M. A., Franks-Skiba, K., Chen, S. & Cooke, R. Myosin ATP turnover rate is a mechanism involved in thermogenesis in resting skeletal muscle fibers. *PNAS* 107, 430–435 (2010).
8. Nag, S. & Trivedi, D. V. To lie or not to lie: Super-relaxing with myosins. *eLife* 10, e63703 (2021).
9. Chu, S., Muretta, J. M. & Thomas, D. D. Direct detection of the myosin super-relaxed state and interacting-heads motif in solution. *Journal of Biological Chemistry* 297, (2021).

Reviewer #3 (Remarks to the Author):

The authors provide no new data to support the mechanistic link between the increased expression of MYLK2 and SRX transformation and reduced ATP consumption. Instead, more literature supporting for their interpretation is included.

The paper by Lewis et al. is now briefly discussed. However, the argument that snap freezing being the reason is highly speculative. Snap freezing should prevent ice crystal formation. Indeed, it can be another issue for the myosin DRX and SRX formation, but this should be confirmed at least with a reference.

Freezing has been demonstrated to significantly impact force production in skinned muscle fibers, directly affecting the active ATP hydrolysis of myosin¹⁻³. Research from established experts in the field, such as Lars Larsson and Carlo Reggiani, who have extensive experience with the skinned fiber model, has consistently shown that freezing reduces tension produced by skinned fibers by approximately 50%. Interestingly, the shortening velocity remains largely unaffected¹. This differential effect can be explained by the underlying mechanisms of muscle contraction:

1. *Shortening velocity is primarily determined by a small number of myosin heads, making it possibly less susceptible to freezing-induced damage.*
2. *Tension development, however, is heavily dependent on:*
 - a) *The number of myosin heads participating in force generation,*
 - b) *The overall structural integrity of the muscle fiber.*

Given these observations, it is highly probable that the freezing and thawing process also affects resting ATPase activity and the equilibrium between the super-relaxed (SRX) and disordered-relaxed (DRX) states of myosin.

Summary:

The manuscript by Napoli, Schmidt et al. presents new information regarding the remodeling of skeletal muscle in hibernating bears. The study is well designed, albeit with a typical caveat that the number of animals is small (9). Nevertheless, multiple measurements for several parameters enhance this reviewer's confidence in the results. Most of the methods are robust, and the results are presented in a clear and informative manner. The conclusions are valid and expand our understanding of the underlying biochemical, biophysical, and anatomical changes occurring in skeletal muscles in bears that hibernate. However, a greater integration of energetics (based on the western results of CI-CV proteins and immunohistochemistry evidence of reduced numbers of mitochondria in hibernation) and expansion to include the results from several earlier muscle studies in bears is requested.

Major Comments:

Materials and Methods

L. 131 - The use of young bears (ages 2-3 years) should be expanded to indicate whether they are sexually mature. This is an important consideration as the hormonal milieu, especially testosterone, could impact skeletal muscle dynamics.

L. 135 - The authors should also specify if the captures during the active season were performed via helicopter darting or trapping.

L. 146 - Please provide a reference for the skinning solution used.

L. 151-170 – Please provide information regarding the controls used for western blot experiments. For example, omission of primary antibody, preabsorption, etc. Also, please indicate the species for primary and secondary antibodies. A reference would also be helpful. Please provide information on the protein stain used in Figure 5B (lower panel, gel stain). Please indicate the dilution used of secondary antibodies.

L. 211-227 – The preparation appears not to consider changes in protein content with season as observed previously (Miyazaki et al., 2022; Hershey et al., 2008; Tinker et al., 1998). Thus, have the authors attempted to normalize the ATPase activity data? This is a similar concern for the proteomic study (L. 247-254).

L. 235 – Please provide a reference for the temperature chosen (17.5C) to perform tension measurements. Was this based on Q_{10} effects or empirical evidence?

L. 292-306 – The preparation of EM tissue would have benefited from a freshly prepared sample process in the same way. The images provided are not very convincing and are somewhat difficult to interpret (see below). How many bears were these taken from? How many from each season?

L. 320 – Please indicate the dilution of secondary antibody used.

Results

L. 344 – Would suggest rewording the heading. “Mild” is used without any context and thus is not informative.

L. 382 – Please provide a definition for “Ktr”.

L. 389-410 – As indicate above, these results would benefit from a normalization either to total mass or total protein content to ensure that the results do not reflect other changes.

L. 468-481 – Have the authors considered quantifying what appears to be a huge difference in mitochondria numbers between seasons (Fig. 4B). If this is not representative, please consider a different image. Additionally, this reviewer is not convinced of the value added by including the EM study as this does not have freshly prepared tissue for comparison. For example, were the number of mitochondria counted and compared to the IHC results? This is actually somewhat surprising given that it would have been available at the time of collection. Suggest removing EM results.

Discussion

L. 506-510 - The discussion would benefit from the addition of comparative numbers revealing the extent of muscle loss in other species, with aging, fasting, inactivity, etc. for comparison.

Throughout - Other than MYLK2, there is a surprising paucity of information (one sentence, L518-520) in the discussion related to the proteomics results. Further, there is little attempt to integrate this information with earlier studies or integration with the biochemical and biophysical results. This integration is required since it represents an important component. For example, were changes in MAFBX (FBX032), MURF1 (TRIM63), or PGC1a observed? These have been proposed previously as mechanisms to reduce the amount of atrophy (Lin et al., 2012).

L. 575-588 – The discussion of MYLK2 and P-MLC2 results overlooks a potential explanation for the maintenance of ATP levels in winter muscle, namely, that the efficiency of phosphorylation (or dephosphorylation) is altered. Specifically, it appears that the ratio of MYLK2 to PMLC2 is much higher in winter than in summer (Fig. 5C). If this is indeed the case (I’m basing this off of estimates from the average), the potential changes phosphatases in addition to kinases could be important players. A search through the proteomics data might provide important clues to the accuracy of this statement.

Minor Comments:

L. 167 – with appropriate

L. 179 – Samples for protein electrophoresis “were”

L. 253-254 – The Hughes reference is incorrectly formatted

L. 259 – Define “FA”

L. 313 – fixated should be “fixed”

L. 382 – please define “Ktr”

L. 445 – mouse is indicated, but no animal protocol is identified for this tissue collection. Under what conditions was this mouse held, strain, etc.?

L. 571 – Use of “comforted” should be “supported”?

L. 654 – If only 9 bears were used in the study, where does the n=11 come from. Presumably this is from counting the two bears that were resampled in year two?

L. 661 – Please provide units for the Y-axis of Fig. 2B. Were these normalized data?

L. 718 – From the Venn diagrams in Suppl. Fig. 3D it appears that only a single protein was present in both summer1 and summer2 muscles. This seems unusual. A similar scenario exists for MYH2 winter1 and winter2. Unless I’m mis-reading these, this suggests poor reproducibility. Please clarify.